# Global phylogeny and taxonomy of *Artemisia*

Bohan Jiao [1,2,3], Meng Wei[1,2,3], Guohao Niu[1,2,3], Xiyang Chen[1,2,3], Yifan Liu [1,2,3], Guangyin Huang[1,2,3], Chen Chen[1,2,3], Jiye Zheng[1,2,3], Jiahao Shen[4], Daniel Vitales[5], Joan Vallès[6], Filip Verloove[7], Andrey S. Erst [8], Alexey P. Seregin [9], Akiko Soejima[10], Xuejun Ge[11], Florian Jabbour [12], Artem Leostrin[13], Goro Kokubugata[14], Wei Wang [1,2,3] & Tiangang Gao [1,2,3] ✉

Developing robust phylogenies and comprehensive taxonomies for big plant genera is crucial for unlocking plant-derived solutions to global sustainability challenges. *Artemisia*, a big genus comprising ~500 species, holds immense medicinal and ecological importance. Despite decades of efforts, establishing a comprehensive phylogeny and taxonomy for global *Artemisia* has remained a formidable challenge. Here, we reconstruct the most comprehensive phylogeny of global *Artemisia* to date (394 species) using a gigamatrix approach. We also analyze evolutionary patterns of 20 morphological characters of *Artemisia* worldwide to evaluate their taxonomic utility. Based on these findings, we propose a global taxonomy for *Artemisia*, recognizing 24 sections in 8 subgenera, and placing 99.6% of accepted species (502/505). This study provides a robust framework to advance understanding of the evolution and ecology of *Artemisia*, and to promote the sustainable utilization of its rich resources. Meanwhile, it introduces an exemplary case for taxonomic research on big genera in the genomic era.

Big plant genera – often defined by thresholds of ≥ 500 species[1,2] – account for ca. 25% of the flowering plant species[2]. Despite their critical role in shaping plant diversity, their global-scale phylogenetic and taxonomic research are significantly behind compared to other genera, owing to their inherent complexity[1,2]. Their phylogeny and taxonomy are uniquely challenging: beyond the sheer number of species, they exhibit extensive (often intercontinental) distributions and frequent rapid radiations (e.g., *Astragalus*[3]), collectively complicating phylogenetic resolution and taxonomic completeness[1,2,4].

The necessary studies were deemed too daunting. This results in a lack of global research on big genera, with the existing studies mostly being regional (e.g., *Myrcia*[5], *Cyperus*[6]). In the past decades, new methods, technologies, and international collaboration have made progress in resolving these challenges, as evidenced by progress in the study of *Solanum*[7] and *Begonia*[8], though the process remains time-intensive. Some big genera are primarily distributed in tropical regions (e.g., *Ipomoea*[4], *Myrcia*[5]), whereas others are in temperate regions (e.g., *Centaurea*[2]). While significant progress has been made

¹State Key Laboratory of Plant Diversity and Specialty Crops, Institute of Botany, Chinese Academy of Sciences, Beijing, China. ²China National Botanical Garden, Beijing, China. ³University of Chinese Academy of Sciences, Beijing, China. ⁴Institute of Botany, Jiangsu Province and Chinese Academy of Sciences, Nanjing, China. ⁵Botanical Institute of Barcelona (IBB, CSIC-Ajuntament de Barcelona), Pg. del Migdia, s.n., Barcelona, Spain. ⁶Laboratori de Botànica - Unitat associada al CSIC - IRBio, Universitat de Barcelona, Av. Joan XXIII 27-31, Barcelona, Catalonia, Spain. ⁷Meise Botanic Garden, Meise, Belgium. ⁸Central Siberian Botanical Garden SB RAS, Novosibirsk, Russia. ⁹Faculty of Biology, M. V. Lomonosov Moscow State University, Moscow, Russia. ¹⁰Faculty of Advanced Science and Technology, Kumamoto University, 2-39-1 Kurokami, Chuo-ku, Kumamoto, Japan. ¹¹Center of Conservation Biology, Core Botanical Gardens, Chinese Academy of Sciences, Guangzhou, China. ¹²Institut de Systématique Evolution Biodiversité (ISYEB), Muséum national d'Histoire naturelle, CNRS, Sorbonne Université, EPHE, Université des Antilles, 57 rue Cuvier CP39, Paris, France. ¹³Herbarium (LE) of Komarov Botanical Institute, Russian Academy of Sciences, St. Petersburg, Russian Federation. ¹⁴Department of Botany, National Museum of Nature and Science, Tsukuba, Ibaraki, Japan. ✉e-mail: Gaotg@ibcas.ac.cn

in resolving a few tropical big genera (e.g., *Ipomoea*[4]), research on temperate ones remains scarce yet presents distinct opportunities. Originating predominantly during mid-Cenozoic geo-climatic upheavals–such as global cooling[9], grassland expansion[10], and desertification[11]–temperate genera often exhibit accelerated diversification rates[12] and acquisition of novel morphological or physiological adaptations to new niches (e.g., cold and/or aridity)[12,13]. These Cenozoic-forged evolutionary innovations position the temperate big genera as unparalleled systems for studying adaptive evolution under environmental upheavals. Their distinct metabolites–such as the antimalarial dominance of *Artemisia annua*'s artemisinin[14], which has replaced tropical *Cinchona*-derived quinine in the past decades–reveal untapped potential for bioprospecting[15]. Consequently, completing the phylogeny and taxonomy of temperate big genera becomes imperative for both evolutionary theory and resources utilization.

Some pioneer attempts have been made to resolve the phylogenies of species-rich taxa by using Sanger sequencing data[16]. However, resolution in these phylogenies is restricted due to limited evolutionary signals in such data. Recent advances in high-throughput DNA sequencing have facilitated the resolution of complex phylogenetic relationships by utilizing extensive evolutionary signals across entire genomes[17]. How to balance taxonomic coverage and genomic depth for a comprehensive and robust phylogeny is a big challenge. The construction of a gigamatrix[18,19], which integrates high-throughput DNA sequencing data (some species and many sequences) with Sanger sequencing data (numerous species and few sequences), offers a promising strategy to address this gap[20].

*Artemisia*, commonly known as wormwood, mugwort, and sagebrush, is a big genus in the family Asteraceae. It comprises over 500 species that are predominantly distributed in the northern temperate regions[21–24]. These species are important in medicine, phytochemistry, and ecological restoration[24] (Fig. 1). As the most famous example, the discovery of artemisinin, a sesquiterpene lactone extracted from *A. annua*[14] (Fig. 1g) for the first time, was recognized with the Nobel Prize in 2015 for being the most effective antimalarial drug to date[25]. This milestone triggered a research boom on the chemistry of *Artemisia* species, with over one thousand papers published annually between 2015 and 2019 (Supplementary Table 1). Recent studies have expanded its therapeutic potential to tuberculosis[26], polycystic ovarian syndrome[27], and allergen-specific immunotherapy via pollen-derived proteins[28]. Ecologically, species like *A. ordosica* and *A. halodendron* play critical roles in desertification control across Asia[29] (Fig. 1e). A high-resolution global phylogenetic framework and a comprehensive infrageneric taxonomy are crucial for understanding and leveraging this scientifically and economically

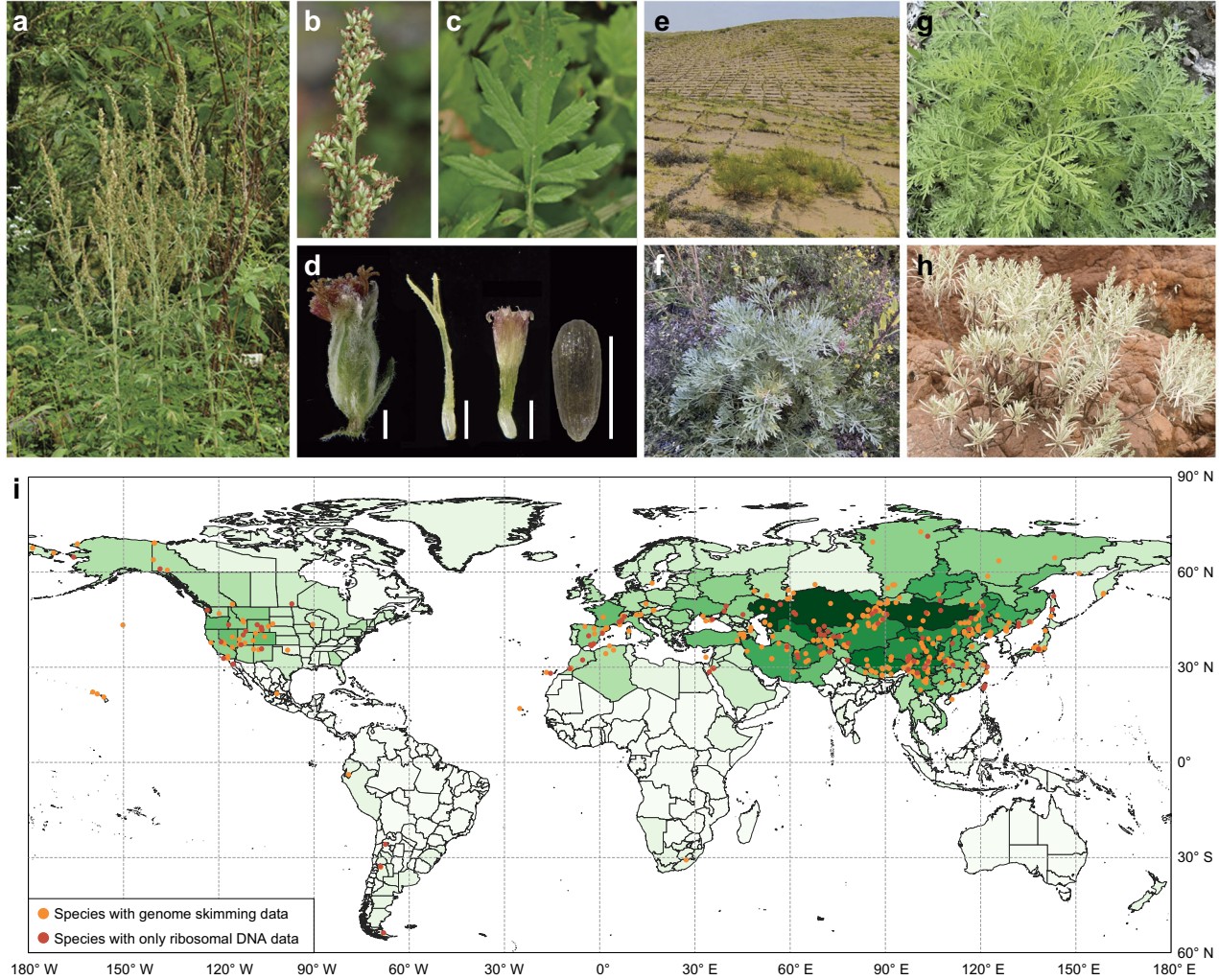

**Fig. 1 | Morphology, distribution and sampling of *Artemisia*. a–d** Morphology of *Artemisia verlotiorum*: **a** Plants; **b** Synflorescence; **c** Leaf; **d** Capitulum, floret and achene, scale bar = 1 mm. **e–h** Economically significant species: **e** *Artemisia oxycephala* (ecological restoration); **f** *A. absinthium* (medicinal); **g** *A. annua* (medicinal); **h** *A. chinensis* (ornamental). **i** Global sampling localities, overlaid on species richness (Global Compositae Checklist[91]; WGSRPD Level 4 ecoregions[92]), the darker the shade of green, the more species are distributed in the region. Sampling point coordinates are listed in Supplementary Data 2.

important genus. But a robust global phylogeny and a complete taxonomic hypothesis for each species of *Artemisia* remain missing, even after decades of efforts (see Supplementary Data 1).

The existing infrageneric taxonomy of *Artemisia* predominantly relies on the morphology of the capitula, leaves, and their life forms[21–23,30]. These morphological characters, however, are prone to repeated evolution, often leading to conflicts with molecular phylogenies or to unreasonable taxonomic treatments[30,31]. For example, the homogamous-discoid capitulum was revealed to evolve independently based on the evidence from cytology and molecular phylogenies[31,32]. Yet all species bearing this state were once classified into a single genus *Seriphidium* based solely on their similar capitulum morphology[33]. Nevertheless, morphological characters remain crucial for taxonomy, identification and evolutionary studies as they show fascinating diversity and are easy to observe and use[34–36]. Morphological variation in numerous taxa, however, remains far from comprehensively studied[34,36], highlighting a big gap in our understanding of their evolutionary trajectories and taxonomic utility. Therefore, it is essential to critically evaluate the already employed morphological characters, actively develop new ones, and identify reliable diagnostic characters for each recognized lineage. Regrettably, this has not yet been done with *Artemisia* on a global scale. Additionally, while some taxonomists[37,38] acknowledged the potential of micromorphological characters in *Artemisia* infrageneric taxonomy, comprehensive studies on them have been scarce[38]. A critical challenge lies in the tiny size of *Artemisia* capitula (mostly < 5 mm in diameter), compounded by the abundance of minute florets within (often > 20 per capitulum) and microscopic internal structures (e.g., anther) inside[21–23].

In this work, we reconstruct a global phylogeny for *Artemisia* with 78% (394 out of 505 species) of its species using a gigamatrix approach by integrating 202 low copy nuclear genes and two ribosomal DNA markers. We also reconstruct phylogenomic frameworks for *Artemisia* utilizing low copy nuclear genes and complete plastomes with 59% of its species respectively, followed by a comparative analysis to clarify the pattern of cytonuclear discordance in *Artemisia*. Furthermore, we analyze 20 morphological characters—traditionally or potentially significant for its infrageneric taxonomy—using herbarium specimens and our field collections across its main distribution ranges (Fig. 1), and infer their evolutionary trajectories. Based on these findings, we propose a global taxonomy for *Artemisia*, including 8 subgenera, 24 sections, and 502 out of the 505 accepted species with taxonomic placements. This study provides a comprehensive framework for advancing our understanding of the evolution, ecology, and sustainable use of *Artemisia*, and introduces a case study on how to tackle taxonomic challenges in big genera in the genomic era.

## Results and discussion

### An expanded phylogenetic framework incorporating 78% *Artemisia* species

We obtained genome-skimming data for 314 species of *Artemisia* and its allies (298 *Artemisia* species), comprising 96 newly sequenced *Artemisia* species not sampled in Jiao et al[31]. (Supplementary Data 2). From these data, we assembled three nuclear datasets: 1) 202 nuclear low-copy genes (NLC dataset); 2) the internal transcribed spacer (ITS) and external transcribed spacer (ETS) regions of nuclear ribosomal DNA (NR dataset); and 3) the concatenated gigamatrix (GM dataset) comprising the NLC dataset and the NR dataset. The final NLC dataset spans 226,916 bp and comprises 314 species (298 ingroups + 16 outgroups), the NR dataset is 916 bp long covering 414 species (394 ingroups + 20 outgroups), and the GM dataset totals 227,832 bp covering 414 species (394 ingroups + 20 outgroups). Among them, 76% (314/414) had both NLC and NR data, while 24% (100/414) were represented solely by NR data. The substantial disparity in data volume between NLC and NR datasets precluded multi-species coalescent analysis (ASTRAL) due to computational infeasibility and statistical

limitation for the gigamatrix[19]. To ensure methodological consistency in phylogenetic analysis, we presented concatenated maximum likelihood (ML) trees throughout the main text.

Using maximum likelihood (ML) analysis of the GM dataset, we reconstructed a fully resolved phylogeny of *Artemisia* (Fig. 2 and Supplementary Fig. 1). Eight clade and 24 subclade nodes showed strong bootstrap support (bootstrap support, BS > 95%), and this tree was completely concordant with the NLC-derived ML tree (Fig. 2, Supplementary Figs. 1 and 2). Topological comparison of all the nuclear phylogenetic trees was shown in Supplementary Fig. 3. Compared with the latest phylogeny (ML tree from the SNP dataset; Supplementary Fig. 3a)[31], the ML trees from GM and NLC datasets increased the species coverage of the *Artemisia* phylogeny from 41% (205/505 species) to 78% (394/505 species, GM dataset; Supplementary Figs. 1 and 3e) and to 59% (298/505 species, NLC dataset, Supplementary Figs. 2 and 3c), respectively. These robust phylogenies enabled the delimitation of 24 strongly supported subclades in 8 clades (Fig. 2 and Supplementary Fig. 3), corresponding to 24 sections in 8 subgenera respectively (detailed in the taxonomy section below and Supplementary Note 1). Our framework resolved evolutionary relationships across 78% of *Artemisia* species with high resolution. Among the 24 subclades, we newly identified 10, revised 11, and confirmed 3 previously established subclades (all BS > 95%). All eight clades showed full support (BS > 95%), consistent with prior work[31]. A comparison of phylogenies between this study and the previous work[31] is summarized in Supplementary Fig. 3. Considering that this study involves taxonomic treatment and phylogeny is a crucial foundation for taxonomy, the terms "subgenera" and "sections" are used simultaneously in the main text to denote the clades and subclades of *Artemisia* respectively, as identified by the GM-derived and NLC-derived ML trees (Fig. 2 and Supplementary Fig. 2).

Some species historically classified within *Artemisia* subg. *Artemisia* based on few morphological characters exhibited unresolved phylogenetic positions in the previous phylogeny based solely on NR dataset[23,39] (Supplementary Fig. 4). Our analyses of GM dataset robustly resolved these species within four distinct subclades (*Auratae*, *Salsoloides*, *Norvegica*, and *Laciniatae*) of *A.* subg. *Dracunculus* (Fig. 2, and Supplementary Figs. 1 and 3), which is further corroborated by following shared morphological characters between them (Supplementary Note 1). For example, *A. phaeolepis*—a species previously placed in *A.* subg. *Artemisia*[23] and unresolved in the ML tree from NR dataset (Supplementary Fig. 4)—is now confidently assigned to the *Laciniatae* subclade of *A.* subg. *Dracunculus* (Fig. 2; Supplementary Note 1). *A. magellanica*, a Patagonian species previously assigned to *A.* subg. *Artemisia*[40] was resolved into *A.* subg. *Pectinatae* in the ML trees of NLC and GM datasets (Fig. 2, and Supplementary Fig.s 1 and 2), with morphological observations aligning congruently with other members of *A.* subg. *Pectinatae* (Supplementary Note 1).

Regarding the circumscription of *Artemisia*, our results supported the inclusion of eight small genera (*Crossostephium*, *Filifolium*, *Kaschgaria*, *Mausolea*, *Neopallasia*, *Picrothamnus*, *Sphaeromeria* and *Turaniphytum*) that had been discussed (BS = 100%; Fig. 2, and Supplementary Fig. 1; see Supplementary Data 1 and 2). Our expanded sampling further revealed that, *Ajaniopsis*, a genus endemic to the Tibet Plateau, was nested in *Artemisia* with strong support[41] (BS = 100%; Supplementary Fig. 1). Furthermore, we classified it into *A.* subg. *Absinthium* sect. *Argyrophyllae* based on the ML tree of NLC and GM dataset (Fig. 2, and Supplementary Figs. 1 and 2). Its unique morphological characters (apically pilose florets, 5- or 6-ribbed achenes, and corymbose synflorescences) may be the result of convergent evolution, as these characters also occur in other species of *Artemisia*—such as *A. albicans* (subg. *Tridentatae*, western North America) and *A. glacialis* (subg. *Absinthium*, European Alps)—that are both phylogenetically and geographically distant[41] (Fig. 2 and Supplementary Fig. 1).

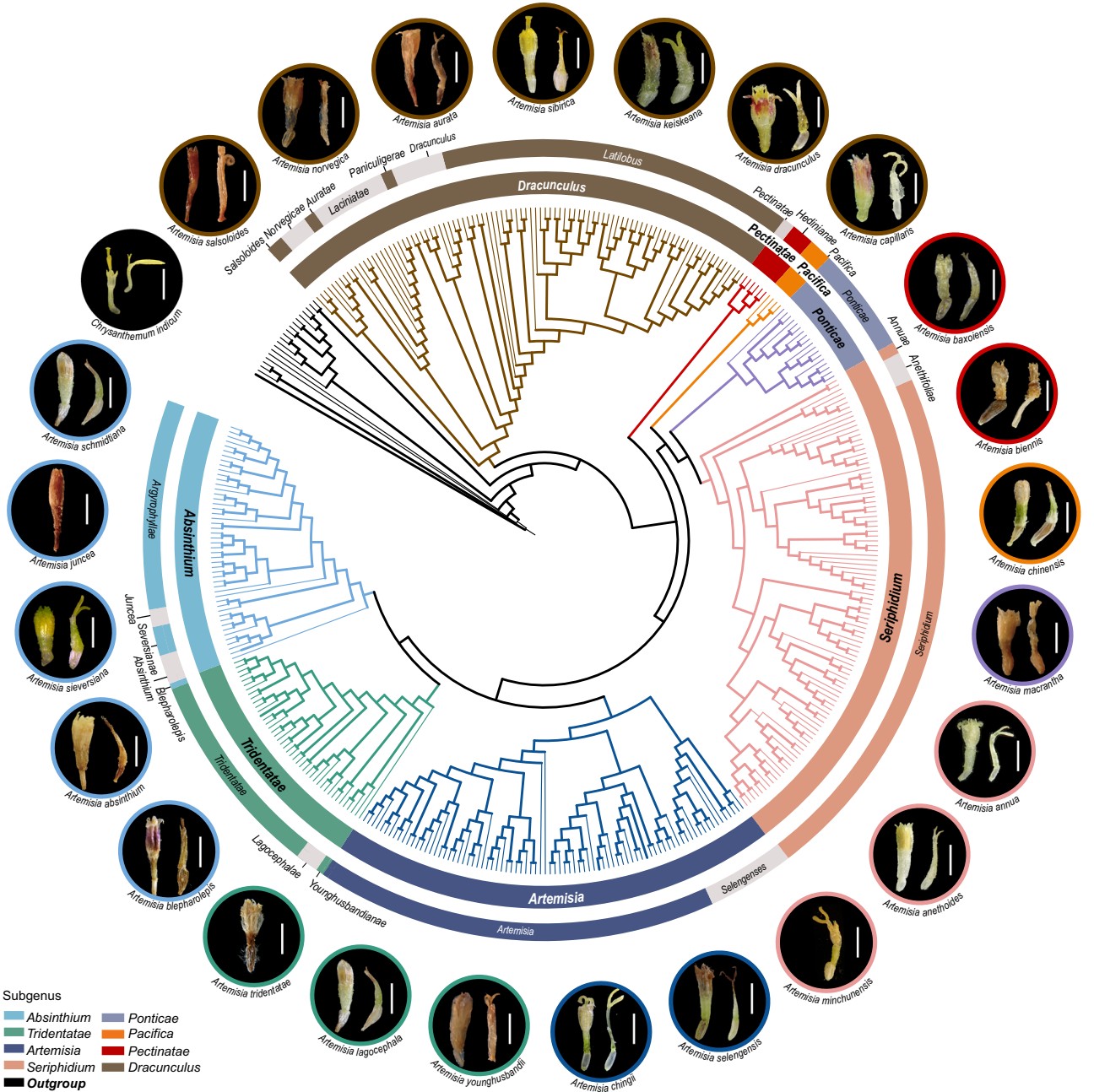

**Fig. 2 | Global phylogeny of *Artemisia*.** Concatenated maximum likelihood (ML) tree of 414 species (394 *Artemisia* + 20 allied species) based on the gigamatrix (GM) dataset. Branches with bootstrap support ≥ 75% are in bold. Colored rings indicate subgenera (inner) and sections (outer); branch colors correspond to subgenera.

Outer images depict floret morphology for each section in *Artemisia* and allied species. The full phylogeny with branch length and detailed support values is provided in Supplementary Fig. 1. Scale bar = 2 mm.

The placement of *Artemisiella* is the only problem remaining for the circumscription of *Artemisia*[31]. The ML trees derived from both GM and NLC dataset strongly supported a sister relationship between *Artemisia* and the clade consisting of *Artemisiella* and *Phaeostigma* (BS = 100%; Supplementary Fig. 3 c and e). In contrast, the ASTRAL tree based on NLC dataset nested *Artemisiella* within *Artemisia*, yet with weak support (local posterior probability, LPP = 0.61; Supplementary Figs. 3b and 5). Our alternative hypothesis tests [Shimodaira-Hasegawa (SH) test[42], Kishino-Hasegawa (KH) test[43], and approximately unbiased (AU) test[44] based on the NLC dataset did not reject the hypothesis of its inclusion in *Artemisia* (Supplementary Table 2). Given that *Artemisiella* can be clearly distinguished from *Artemisia* by morphological

characters (e.g., leaves oblong, 3-pinnatisect, with 8–13 pairs of lateral lobes)[45], we tentatively treated it as a separate genus.

## Cytonuclear discordance and possible hybridization within *Artemisia*

We assembled complete plastomes for 314 species (298 *Artemisia* + 16 allied species, plastome dataset) from our genome-skimming data to assess cytonuclear discordance and possible hybridizations in *Artemisia*. This 113,792 bp dataset (80 coding regions, inverted repeats excluded) yielded a robust phylogeny (78% nodes BS > 95%; Supplementary Fig. 6), revealing extensive cytonuclear discordance in *Artemisia* (Fig. 3a; full topologies in Supplementary Figs. 2, 4, and

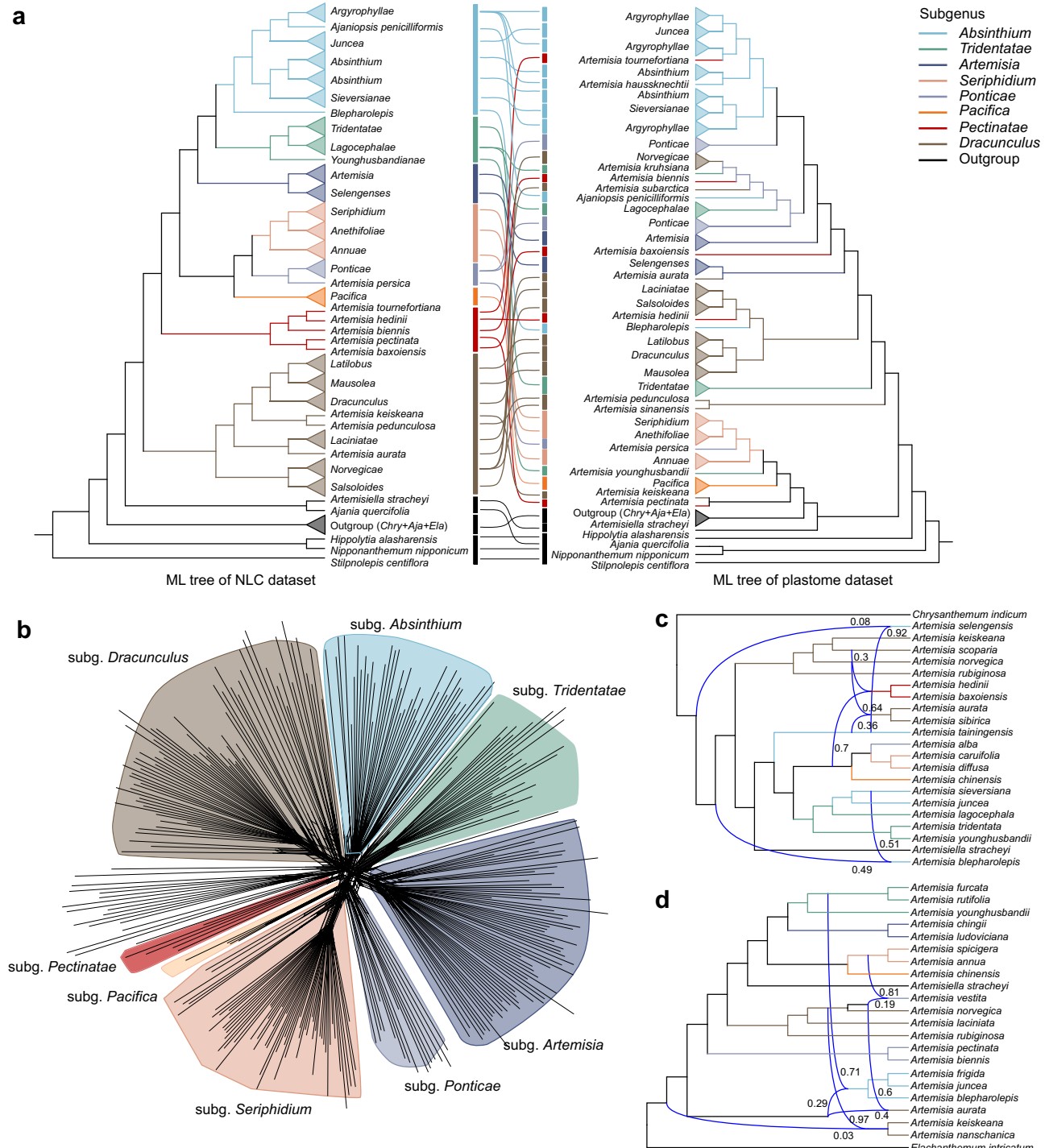

**Fig. 3 | Phylogenetic discordance and reticulate evolution in *Artemisia*.**
**a** Tanglegram comparing maximum likelihood (ML) phylogenies from 202 low-copy nuclear genes (NLC dataset, left) and plastome dataset (right), with connecting lines matching sections (color-coded by subgenus). The full phylogeny with branch length and detailed support values is provided in Supplementary

Figs. 2 and 6. **b** Split network based on uncorrected *p*-distances NLC dataset. **c**, **d** Optimal phylogenetic networks reconstructed from two different 22-taxa NLC datasets with maximum number of reticulations ($h_{max}$) = 4. Curved branches denote hybrid nodes annotated with inheritance probabilities; branch colors correspond to subgenera.

6–8). The plastome phylogeny showed that *Artemisia* was not monophyletic, with *Chrysanthemum-Ajania-Elachanthemum* clade and *Artemisiella* clade nested within it (Fig. 3a and Supplementary Fig. 6, BS = 82%, 96%). And seven of the eight clades (subgenera) are polyphyletic, except for *A.* subg. *Pacifica* (Fig. 3a), which has only four species. This cytonuclear discordance suggested the possibility of rapid evolutionary radiations or hybridizations within specific lineages of *Artemisia*. To identify potential causes, we quantified

genealogical concordance[46], conducted polytomy tests[47], and performed phylogenetic network analyses[48] based on the NLC dataset (see "Methods"). We measured phylogenomic genealogical discordance using gene concordance factors (gCF) derived from the NLC dataset (Supplementary Fig. 2). This factor evaluates the percentage of gene trees that agree with various nodes on the phylogenetic tree and low gCF values (< 5%) may arise from insufficient information (such as short branches) or from genuine conflicting

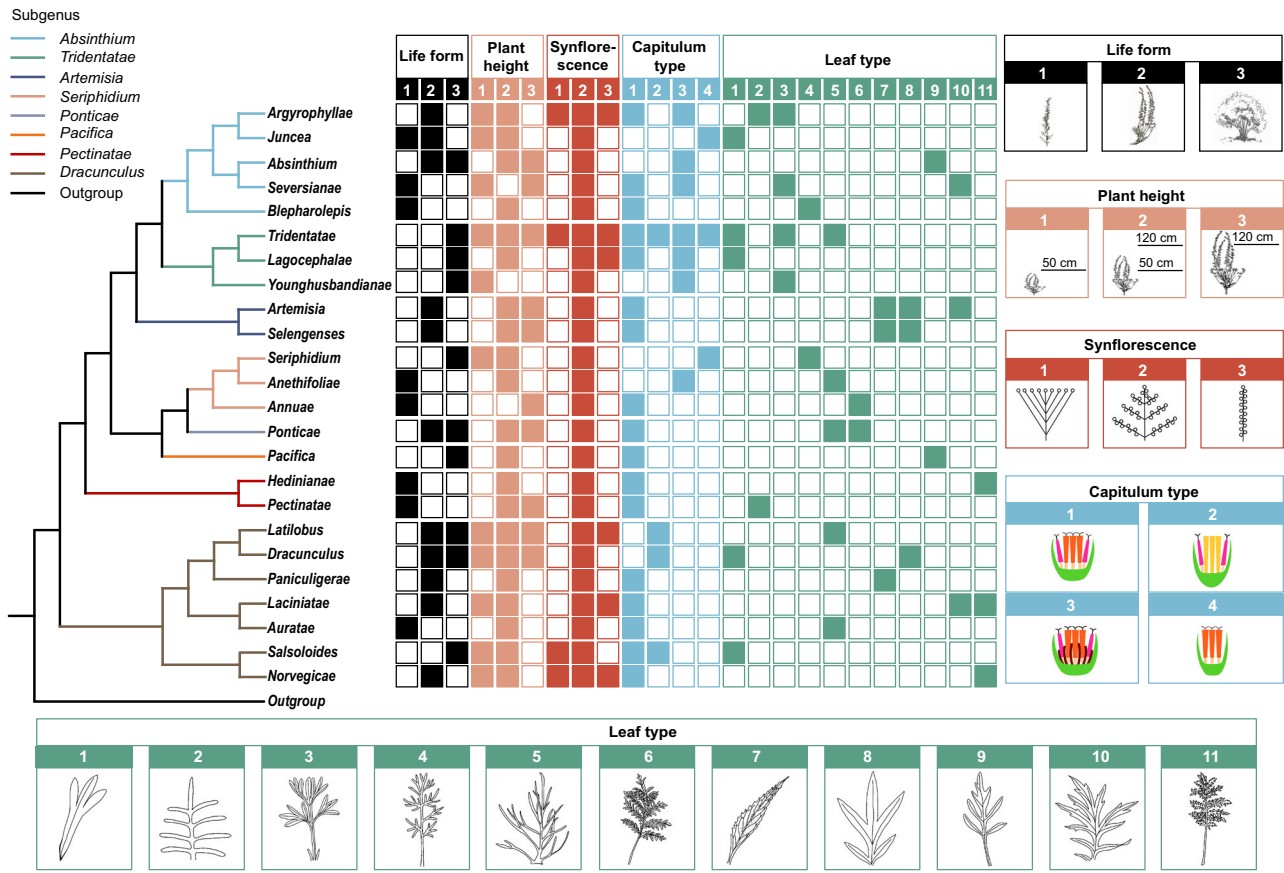

**Fig. 4 | Macromorphological character variation across 24 sections of *Artemisia*.** Schematic maximum likelihood (ML) tree topology based on the gigamatrix (GM) dataset shown on the left, with branch colors indicating subgenera; selected macromorphological characters (e.g., leaf type) and their states depicted on the right and bottom. Detailed character state definitions are provided in Supplementary Data 3.

signals[46]. The ML tree of NLC dataset exhibited high bootstrap support (mean value = 89.50%; Supplementary Fig. 2) and low gCF (mean value = 3.66%; Supplementary Fig. 2), implying conflicting signals among genes. The polytomy tests retained the null hypothesis of zero branch lengths (indicating unresolved relationships or polytomies rather than bifurcating nodes) for all inter-subgeneric nodes ($p > 0.1$; Supplementary Fig. 5), except the divergence node of *A.* subg. *Pacifica* ($p = 0$; Supplementary Fig. 5). Meanwhile, 79% of the nodes within subclades (sections) and among species could not rule out the polytomy hypothesis ($p > 0.1$; Supplementary Fig. 5). These results extended beyond known radiated subgenera (*A.* subg. *Dracunculus*, 81% nodes $p > 0.1$; *A.* subg. *Seriphidium*, 87% nodes $p > 0.1$)[39] to the third-biggest subgenus *A.* subg. *Artermisia* (91% nodes $p > 0.1$). By applying phylogenomic data, we preliminarily ruled out the possibility of unresolved relationships in polytomy tests, and inferred that the evolutionary history of *Artemisia* might have been persistently accompanied by incomplete lineage sorting (ILS) resulting from rapid diversification, which could be a cause of low gCF. Rapid diversification has thereby complicated phylogenetic resolution at the subgeneric and species levels[39,49,50] and necessitating phylogenomic data for robust inference of subgeneric and species-level relationship in big genera.

Hybridization within *Artemisia* has long been hypothesized[51-53], yet lacked rigorous global-scale testing. We analyzed two simplified 22-species datasets (20 *Artemisia* species representing all subgenera + 2 outgroups including *Artemisiella*; Fig. 3c and d), given the computational limitation of PhyloNet v.3.6.9. The results revealed widespread hybridization during the early evolution of *Artemisia* (Fig. 3c, d, and

Supplementary Figs. 9 and 10), and plausibly explained the close phylogenetic relationships observed between species from different subgenera in the plastome tree (Fig. 3a and Supplementary Fig. 6). For example, *A.* subg. *Dracunculus* exhibited obvious reticulate evolution in the Split network, especially with its close relatives *A.* subg. *Pectinatae* and *A.* subg. *Absinthium* (Fig. 3b). The plastome phylogeny showed two species, i.e., *A. hedinii* (*A.* subg. *Pectinatae*) and *A. blepharolepis* (*A.* subg. *Absinthium*), were embedded within *A.* subg. *Dracunculus* (BS = 100%; Fig. 3a). PhyloNet analysis also indicated that members of *A.* subg. *Dracunculus* was involved in the hybrid origin of members of *A.* subg. *Pectinatae* ($\gamma = 0.3$; Fig. 3c), and its ancestral lineages participated in the hybrid origin of *A. blepharolepis* ($\gamma = 0.49$; Fig. 3c). Morphologically, *A. blepharolepis* can be distinguished from other species of *A.* subg. *Absinthium* by having 5 - 8 leaf segment pairs (vs. < 4 pairs in other species of *A.* subg. *Absinthium*)[23]. Interestingly, this character closely resembled that of members of *Laciniatae* clade (*A.* subg. *Dracunculus*), its sister in the plastome tree (Fig. 4). Additionally, *A. pectinata* (*A.* subg. *Pectinatae*) has been observed to share sterile disk florets with some lineages of *A.* subg. *Dracunculus*[23], providing additional evidence for their potential hybrid origin.

Chloroplast capture likely contributed to cytonuclear discordance in *Artemisia*. Species of *A.* subg. *Pectinatae* were scattered across the plastome tree, with five sampled species nested within five distinct subgenera (Fig. 3a). Conversely, nuclear phylogeny strongly supported their monophyly (BS = 100%; Fig. 3a), despite minimal gene tree concordance (gCF = 0.99%; Supplementary Fig. 2). This conflict—nuclear cohesion versus chloroplast dispersion—aligned with chloroplast capture dynamics[54]. These five species and their plastome-

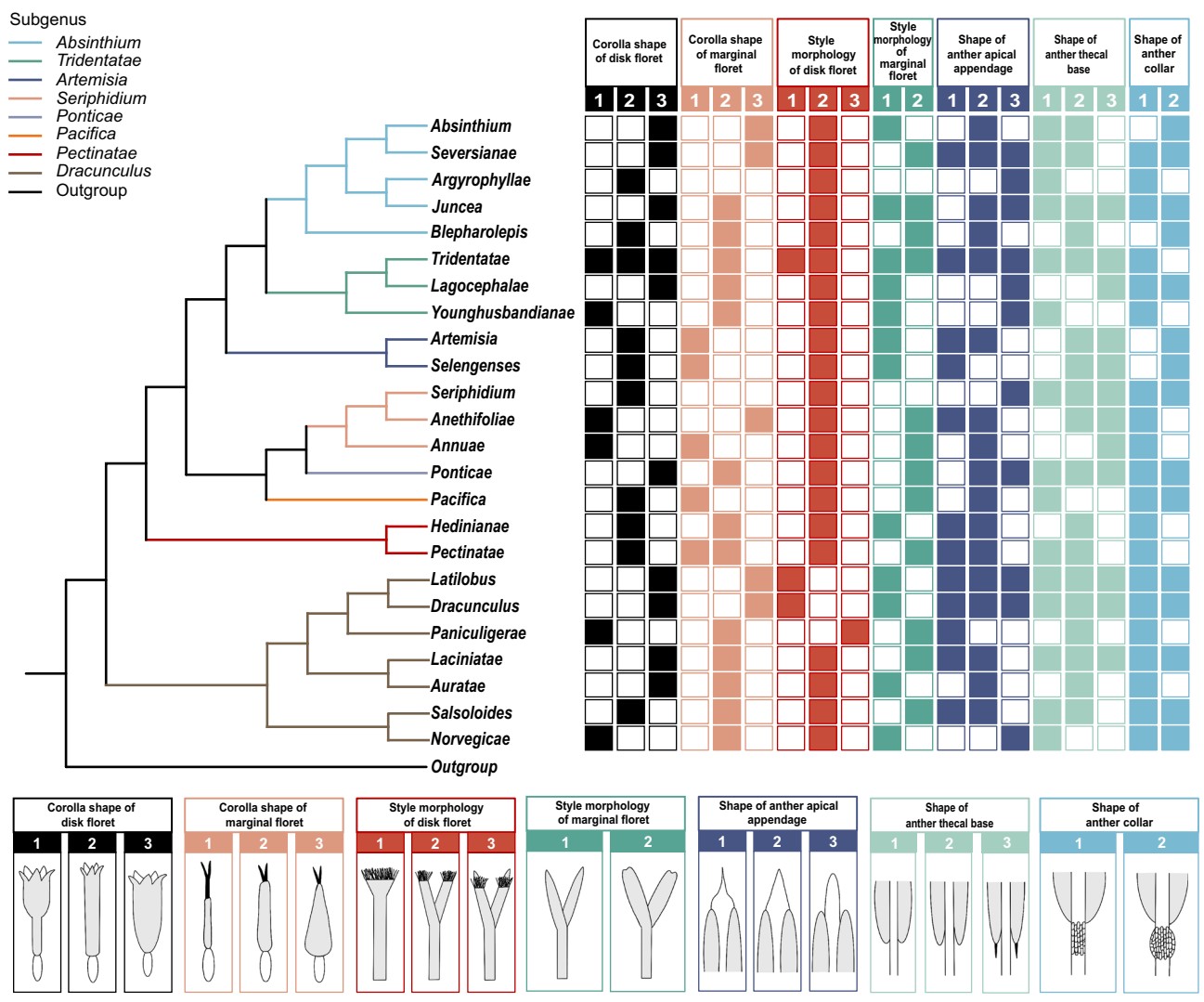

**Fig. 5 | Micromorphological character variation across 24 sections of *Artemisia*.** Schematic maximum likelihood (ML) tree topology based on the gigamatrix (GM) dataset shown on the left, with branch colors indicating subgenera; seven micromorphological characters and their states depicted on the right and bottom. Detailed character state definitions are provided in Supplementary Data 5.

implied relatives showed sympatric/parapatric distributions (Supplementary Figs. 6 and 11), providing spatial evidence for the possible chloroplast capture.

In summary, our results show hybridization and chloroplast capture have occurred within specific *Artemisia* lineages. These events lead to cytonuclear discordance, highlighting the limitations of relying solely on maternally inherited plastome data to reconstruct its evolutionary history. In particular, chloroplast capture events can mislead phylogenetic inference, further complicating the interpretation of organismal relationships[55]. Consequently, the current taxonomy of *Artemisia* primarily relies on nuclear-based phylogenies, which provide a more robust framework for resolving relationships[56]. Nevertheless, we recognize plastome remains an essential component of plant genomes. Conducting comparative phylogenetic analyses that incorporate plastome data can yield valuable insights into the evolutionary journey of *Artemisia*. In this way, it serves as a complementary approach to the nuclear-based phylogenies[57].

### Evolutionary trajectories of morphological characters

We totally studied 20 (13 macro- and 7 micro-) morphological characters that can potentially be used for *Artemisia* infrageneric taxonomy by reconstructing their ancestral states (Figs. 4, 5, and Supplementary Figs. 12–14), and testing their phylogenetic signals (see details in "Methods"; Supplementary Data 3–5).

Thirteen macromorphological characters, including life form, plant height, synflorescence, capitulum, and leaf morphology (Supplementary Tables 3 and 4) were analyzed based on the ML tree derived from GM dataset (Fig. 4 and Supplementary Fig. 12). Four of them (capitulum type, leaf type, leaf segment width and leaf area) exhibited strong ($\lambda \geq 0.9$ or $K \geq 1.5$) or moderate ($0.9 > \lambda \geq 0.7$ or $1.5 > K \geq 1$) phylogenetic signals (Fig. 4 and Supplementary Fig. 12, Supplementary Data 3), and thus could be useful in delineating subgenera and sections within *Artemisia*. The other characters (mostly vegetative) show weak ($\lambda < 0.7$ or $K < 1.5$) phylogenetic signals (Supplementary Data 3) and are therefore not ideal for infrageneric taxonomy.

Notably, aside from capitulum type, which was previously employed in infrageneric taxonomy[21–23] ($\lambda = 0.93$, $K = 1.99$; Fig. 4 and Supplementary Fig. 12, Supplementary Data 1 and 3), leaf type, a character introduced in this study, was identified as a key infrageneric taxonomic character due to its strong phylogenetic signal ($\lambda = 0.99$, $K = 1.50$; Supplementary Fig. 12, Supplementary Table 3, Supplementary Data 3). For example, with the help of leaf type, we can further identify two new sections in the recently established subgenus *A*. subg.

*Pectinatae*[31]. Species of sect. *Pectinatae* have Type 2 leaves (1-pinnatisect multiple-lobed, small), while species of sect. *Hedinianae* have Type 11 leaves (2-pectinately pinnatisect, large) (Fig. 4, Supplementary Table 3). Besides, leaf type is more readily observable than capitulum type (Fig. 4), which usually requires dissection under a stereomicroscope for identification. Other leaf morphological characters, like leaf segment width and number of leaf segment pairs, are often used for the subgeneric taxonomy and species identification in *Artemisia* (Supplementary Data 1). Our results showed that, except for the leaf segment width ($\lambda = 0.73$, $K = 1.05$; Supplementary Fig. 12, and Supplementary Data 3) and leaf area ($\lambda = 0.81$, $K = 1.10$; Supplementary Fig. 12, and Supplementary Data 3) which exhibit moderate phylogenetic signals, most others showed weak phylogenetic signals ($\lambda < 0.7$ or $K < 1$; Supplementary Data 3) and are therefore not useful for infrageneric taxonomy.

We also studied seven micromorphological characters in florets across 200 *Artemisia* species based on the pruned GM-derived ML tree, including the corolla shape and style morphology of both marginal and disk florets, along with the shape of the anther's apical appendage, thecal base, and collar in disk florets (Fig. 5, and Supplementary Figs. 13 and 14; Supplementary Table 5; Supplementary Data 5). Although some taxonomists[37,38] realized the taxonomic potential of these characters, to our knowledge, none have been systematically studied[38,58]. This study initiated a global investigation, addressing a knowledge gap.

Of the seven micromorphological characters, four of them (e.g., corolla shape and style morphology of both marginal and disk floret) displayed strong phylogenetic signals, which could assist in delimiting subgenera and sections (Fig. 5). Notable examples include corolla shape of disk floret ($\lambda = 0.99$, $K = 0.97$), corolla shape of marginal floret ($\lambda = 0.99$, $K = 3.22$), style morphology of disk floret ($\lambda = 0.99$, $K = 4.41$), and style morphology of marginal floret ($\lambda = 0.99$, $K = 2.50$) (all in Supplementary Data 5). While anther-related characters exhibited weak phylogenetic signals ($\lambda < 0.7$ or $K < 1$; Supplementary Data 5), our observations revealed stable interspecific variations in anther apical appendage and thecal base morphology (Supplementary Figs. 13 and 14). Thus, it would be beneficial to expand sampling and test the value of these characters in species delimitation.

Our studies also revealed some interesting evolutionary trajectories of these morphological characters. For example, homogamousdiscoid capitulum (*Seriphidium* type), a character state in *Artemisia* (Fig. 4 and Supplementary Fig. 12) was once used to circumscribe the genus *Seriphidium*[33]. Our inferred phylogeny (Fig. 2) and morphological analysis (Fig. 4) revealed that it evolved independently at least three times in Asia, North America and North Africa (Fig. 4 and Supplementary Fig. 12)—regions where species with this state inhabit arid zones across the three continents[21–23]. It seems that *Artemisia* lineages in these geographically isolated regions are repeatedly playing the same tape of life[59].

All these efforts and results indicated that morphology, as the most direct evidence in taxonomic research, remained worthy of indepth investigation even in the genomic era[34–36].

### The comprehensive sectional taxonomy of *Artemisia*
Following the principles of monophyly and identifiability[35], we proposed a complete sectional taxonomy of *Artemisia*, identified diagnostic morphological characters for each subgenus and section, and developed an identification key to all of them (Supplementary Note 1) based on the phylogenies from genomic data (Fig. 2) and analysis of morphological characters (Figs. 4 and 5). This update included the establishment of 10 new sections and revised circumscriptions for 11 existing ones (Supplementary Note 1). The species that were not sampled in the GM-derived ML tree were assigned to a subgenus and section by employing the diagnostic morphological characters revealed in the current study. In total, 502 of 505 accepted *Artemisia* species were classified into 8 subgenera and 24 sections, 394 species

were assigned based on both molecular and morphological evidence, while 105 species solely on diagnostic morphological characters (Supplementary Data 6). Three species (*Artemisia avarica*, *A. dipsacea*, *A. galinae*) remained unplaced due to insufficient morphological and molecular evidence, primarily stemming from ambiguous protologues and specimen inaccessibility. Most *Artemisia* species now have testable hypotheses for their subgeneric and sectional placements (Supplementary Data 6). The differences between the present taxonomy and previous taxonomies (most regional) were summarized in Supplementary Data 6.

The current taxonomic framework decomposes this big and complex genus (505 species) into 24 smaller sections (mean 19 species/section; range 1–119). Most sections are confined to a single country or region, enabling local taxonomists to handle them within a practical timeframe. Additionally, since each section has distinct morphological characters (Supplementary Note 1), this framework will not only aid in identification but also encourage local taxonomists—especially those without access to molecular facilities—to revise the taxonomy of *Artemisia* in their own regions based on morphology. Data obtained from these local revisions could then be continuously integrated back into the present framework, progressively enhancing species and geographic coverage toward a comprehensive monograph. The present taxonomic framework thus facilitates crosscontinental collaboration and data integration, exemplifying a generalizable workflow for updating phylogenies and taxonomies of big genera in the genomic era (Supplementary Fig. 15). Unplaced species and those unsampled in molecular phylogeny warrant further sampling, sequencing, and integration. Notably, our single-representativeper-species approach does not address species delimitations in *Artemisia*, and thus, this species treatment herein remains provisional, consistent with the inherently dynamic nature of taxonomic studies.

### Summary
Global-scale phylogenetic and taxonomic studies on big plant genera have been metaphorically termed a "black hole" due to their inherent complexity[4,35]. Our integrative analysis of genomic and morphological data resolves the long-standing phylogenetic and taxonomic complexities in the big genus *Artemisia*, establishing a phylogenetically robust and comprehensive taxonomic framework that proposed testable hypotheses for nearly all recognized species. Our study, along with other large-scale investigations of tropical big genera (e.g., *Ipomoea*[4]), demonstrates that actively adopting new technologies, exploring novel morphological characters, and fostering global collaboration, can tackle these long-standing challenges, and even accelerate the process that might otherwise take decades or even centuries. Our study provides a robust baseline for future studies of this ecologically and economically important big genus, such as local revisions, bioprospecting, conservation, and comparative evolutionary analysis. Future research could prioritize local revisions within this framework, enhanced by increasing sampling density and integrating genomic data with fine-scale morphological, geographical and ecological data. This approach is crucial for clarifying species boundaries in *Artemisia*, particularly within notoriously difficult groups such as the *A. vulgaris* complex. This work serves as an exemplary case for taxonomic research on big genera in the genomic era.

### Methods
#### Taxon sampling, DNA extraction, PCR reaction and sequencing
We sampled 394 *Artemisia* species spanning all eight subgenera of *Artemisia* currently recognized[31]. This sampling included all genera now classified within *Artemisia* but formerly placed in segregate genera—specifically *Ajaniopsis*, *Crossostephium*, *Filifolium*, *Kaschgaria*, *Mausolea*, *Neopallasia*, *Picrothamnus*, *Sphaeromeria*, and *Turaniphytum*—based on the most recent phylogeny[31] (Supplementary Data 2). We also sampled 20 species from its closely related genera within the tribe Anthemideae as outgroups based on the

recent phylogeny of Anthemideae[60,61], including *Artemisiella, Chrysanthemum, Ajania, Phaeostigma, Elachanthemum, Stilpnolepis, Nipponanthemum, Tanacetum, Brachanthemum, Hippolytia,* and *Allardia*. Each species was represented by a single sample. Of the 414 species (ingroups+outgroups) analysed, 314 species had genome-skimming data (96 newly sequenced species), while the remaining 100 species had only nuclear ribosomal DNA data (ITS, ETS). Of these 100, 18 were generated in this study and 82 were retrieved from GenBank (Supplementary Data 2). From the genome-skimming data, four datasets were generated: 1) low-copy nuclear gene sequences (NLC dataset), 2) nuclear ribosomal DNA sequences (NR dataset), 3) the concatenated gigamatrix (GM dataset) comprising the concatenated NLC dataset and the NR dataset, and 4) complete plastome (plastome data). The sampling details for each subgenus of *Artemisia* are as follows: 1) *A.* subg. *Dracunculus*, comprising 94 out of the total 122 species, with genomic data available for 73 species; 2) *A.* subg. *Pectinatae*, which includes six out of 10 species, with genomic data accessible for five species; 3) *A.* subg. *Pacifica*, encompassing all four species, with genomic data available for all species; 4) *A.* subg. *Ponticae*, comprising 17 out of 23 species, with genomic data accessible for 14 species; 5) *A.* subg. *Seriphidium*, which includes 96 out of the total 128 species, with genomic data available for 65 species; 6) *A.* subg. *Absinthium*, comprising 50 out of the total 63 species, with genomic data accessible for 39 species; 7) *A.* subg. *Tridentatae*, which includes 41 out of 41 species, with genomic data available for 29 species; and finally, 8) *A.* subg. *Artemisia*, which comprises 86 out of the total 111 species, with genomic data accessible for 69 species. Supplementary Data 2 provides detailed sampling information.

Total genomic DNA was extracted using the TIANGEN plant genomic DNA extraction kit (TIAN-GEN Biotech., Beijing, China) following the manufacturer's protocol. Total DNA extracted from silica-gel-dried leaves was sheared into ~350 bp fragments to build 350 bp insert libraries, and unsheared DNA from herbarium specimens was used to construct 150 bp insert libraries. The DNA libraries were constructed using the NexteraXT DNA Library Preparation Kit (Illumina, Shanghai, China) and were sequenced on the Illumina HiSeq Xten platform (Illumina, Shanghai, China). We obtained ~3 Gb of data for each sample with paired-end libraries. The average read length was 150 bp for silica-gel-dried leaves, and 100 bp for herbarium specimens.

## Transcriptome assembly and low-copy nuclear gene selection

We first selected low-copy nuclear genes from transcriptome sequencing data of nine diploid *Artemisia* species representing all eight subgenera and one outgroup (Supplementary Data 2). Ploidy was cytologically confirmed (Supplementary Method 1; Supplementary Fig. 16). Then we extracted the target nuclear genes from the genome-skimming data. RNAs were isolated from leaves and/or floral buds using the ZR Plant RNA MiniPrep kit (Zymo Research, Orange, CA, USA) following the manufacturer's protocol. Library preparation utilized the PureLink RNA Mini Kit (Invitrogen, Carlsbad, CA, USA), and sequencing was performed on the Illumina HiSeq 2500 platform, employing 100-150 bp paired-end reads with a 6 Gb throughput. Raw RNA-seq reads were cleaned using SeqyClean v1.10.09[62] to trim poly-A/T tails and terminal nucleotides with a 10 bp sliding window (Phred score ≤10). The filtered reads were de novo assembled with Trinity v2013-11-10[63], yielding an average of 145,547 transcripts per sample. To estimate orthologs counts, we applied Yang and Smith's pipeline[64] using the rooted ingroups (RT) method. We ran MarkerMiner v1.2[65] using *Artemisia annua* as a reference and with a minimum transcript length of 400 bp. Following Vargas et al[66]., we used GoldFinder (https://bitbucket.org/oscarvargash/goldfinder)[66] to efficiently sub-select markers from the results of MarkerMiner v1.2[65] to select the best markers, and identified 253 single-copy nuclear genes as target nuclear gene markers with an average length of 1692 base pairs (bp) per marker.

## Obtaining the target markers from genome skimming data

Following Vargas et al[66]., we used GoldFinder selected markers as references to retrieve target markers from genome-skimming data. First, we used SeqyClean v1.10.09[62] to clean the genome-skimming data using the same parameters as for transcriptome data. Then, we used the script bbmap.sh (https://sourceforge.net/projects/bbmap/) to filter out reads from chloroplast, mitochondrial, and ribosomal DNA by mapping the genome-skimming data to the references using default parameters, retaining only reads from the nuclear genome. The reference genomes used are the chloroplast genome of *Artemisia frigida* (GenBank accession: NC020607), the mitochondrial genome of *Chrysanthemum indicum* (GenBank accession: MH716014), and the ribosomal DNA of *Eschweilera congestiflora* (GenBank accession: JN222324, JN222317). For each of 253 target low copy nuclear markers, the longest sequence in the marker matrices was chosen as the reference using the script longest_seq_fasta.py[66]. The genome skimming data was aligned to the reference sequences using the script sam2-consense.py (https://github.com/edgardomortiz/sam2consensus), in order to generate consensus sequences. The script baits_file_organizer.py was used to separate the sequences of each sample into single files, and then merge the homologous genes from all samples into a matrix using cat_fastas_per_gene.py[66]. Multiple sequence alignments were performed using prank_wrapper.py[64], followed by alignment trimming with phyutility_wrapper.py[64]. Given the shallow sequencing depth of the 3 Gb genome skimming data, and to ensure species coverage and phylogenetic information of the final obtained markers, only 202 markers (with 50% species coverage and matrix length >150 bp) were retained from the 253 markers. These were concatenated using concatenate_matrices.py[64] for downstream phylogenetic analysis.

## Plastome assembly and annotation

We de novo assembled complete plastomes for 314 species (298 *Artemisia* + 16 allied species) using GetOrganelle v1.7.6[67] and recommended parameters (Supplementary Data 2). Assembled plastomes were annotated using GeSeq v2.03[68] (CHLOROBOX; https://chlorobox.mpimp-golm.mpg.de/index.html), with the *Artemisia frigida* plastome (GenBank accession: NC020607) as a reference. tRNA annotation was performed using tRNAscan-SE v2.0.733[69]. Annotation results were imported into Geneious v. 11.0.4[70], where the amino acid sequences of protein-coding regions were validated and annotations adjusted accordingly. Based on the adjusted annotations, coding and non-coding regions were extracted using a Python script get_annotated_regions_from_gb.py (https://github.com/Kinggerm/PersonalUtilities/blob/master/). Each region was aligned using MAFFT v7.2234[71] and adjusted manually using BioEdit v.7.0.5.3[72]. This generated three datasets: 1) a concatenated matrix of the entire plastome (including all coding and non-coding regions); 2) a CDS matrix concatenated from all protein-coding regions; 3) a non-CDS matrix concatenated from all non-coding regions (Supplementary Data 2).

## Ribosomal DNA sequence assembly and acquisition

We de novo assembled nuclear ribosomal DNA (ITS and ETS) sequences of 314 *Artemisia* species from genome-skimming data using GetOrganelle v1.7.6[67] with the recommended parameters (Supplementary Data 2). Additionally, we generated ITS/ETS sequences for 18 *Artemisia* species following the protocol of Jiao et al[58]. (detailed in Supplementary Method 2) and retrieved those of 82 congeneric species from GenBank, yielding a final NR dataset of 414 species (394 *Artemisia* + 20 allied species).

## Gigamatrix approach

We generated a gigamatrix (GM dataset) of 414 species by concatenating the NLC dataset (202 nuclear low-copy genes from 314 species via genome-skimming data) and NR dataset (ITS and ETS sequences for 414 species, including 314 from genome-skimming, 18

newly generated in this study, and 82 from GenBank; see Supplementary Data 2). The gigamatrix approach offers two key advantages: (1) its higher-level relationships align closely with those in phylogenomic trees, and (2) it enables accurate placement of some species using only ITS and ETS sequences[19]. However, several limitations should be noted: 1) Data heterogeneity impacts: different evolutionary rates between NR data and NLC data may introduce biases in branch length estimation, necessitating partitioned models for correction. 2) Low support for certain taxa: taxa relying solely on a few markers (e.g., ITS/ETS) may have unstable placements and weak support due to high missing data rate. 3) Constraints on species-tree methods: Non-random distribution of missing data (e.g., incomplete NLC coverage across taxa) may reduce the reliability of multispecies coalescent methods (e.g., ASTRAL)[19]. Thus, we used IQ-TREE v.2.0.6[73] to build maximum likelihood (ML) trees for the gigamatrix, treating each nuclear gene and ribosomal DNA markers as a separate partition. Substitution models were selected via the corrected Akaike information criterion (AICc) calculated using ModelFinder in IQ-TREE.

### Tree topology tests

The topologies generated from plastome dataset and NLC dataset (inferred via maximum likelihood and coalescent methods) revealed incongruent phylogenetic relationships between *Artemisia* and its allies. To find an optimal tree from these genomic datasets, we compared the likelihood values of alternative hypotheses against those of the unconstrained ML tree using three statistical tests generated in IQ-TREE v.2.0.6[73]: Shimodaira-Hasegawa (SH) test[42], Kishino-Hasegawa (KH) test[43], and the approximately unbiased (AU) test[44]. Constrained trees were constructed in Mesquite v3.61[74] by enforcing monophyly on target taxa (with others as polytomies), representing three scenarios: 1) monophyly of *Artemisia*; 2) *Artemisia* + *Artemisiella* as a clade; and 3) *Artemisia* + *Artemisiella* + (*Chrysanthemum-Ajania-Elachanthemum*) as a clade. These constrained topologies were then optimized in IQ-TREE under the GTR + I + G model (partitioned by DNA regions) and used to test alternative hypotheses of tree topology.

### Analysis of genomic data and causes of cytonuclear discordance

We built two datasets for phylogenetic reconstruction: 1) NLC dataset; 2) plastome dataset. For the NLC dataset, phylogenetic analysis was performed using two methods. First, we used IQ-TREE v.2.0.6[73] to build a ML tree for the concatenated matrix. Substitution models were selected via the corrected Akaike information criterion (AICc) calculated by ModelFinder[75] in IQ-TREE. After building the ML tree, we measured phylogenomic discordance using gene concordance factors (gCF) and site concordance factors (sCF) calculated in IQ-TREE[46]. These factors evaluate the percentage of gene trees that agree with various nodes on the phylogenetic tree and quantify the number of informative sites that support different topologies. Low gCF values may arise from insufficient information (e.g., short branches) or from genuine conflicting signals. Similarly, low sCF values (~30%) suggest a lack of phylogenetic information in the loci[46]. Second, we applied the multispecies coalescent method with ASTRAL III v.5.7.4[76]. The input gene trees were generated using IQ-TREE. Before running ASTRAL III, we collapsed branches with bootstrap support (BS) ≤ 20% in gene trees using the 'nw_ed' function in Newick Utilities v1.6[77]. The coalescent-based species tree was reconstructed with ASTRAL-III, using gene trees generated for each single-gene matrix with IQ-TREE as input. Polytomy tests were conducted in ASTRAL-III, using the ASTRAL-III topology (option -t 10)[47]. Polytomy test is a statistical test, for the null hypothesis that a branch of a species tree is a polytomy given a set of gene trees, which helps identify the presence of polytomies while considering incomplete lineage sorting (ILS). A *p*-value ≥ 0.1 indicates that the null hypothesis cannot be rejected. We chose this conservative *p* value because

phylogenetic analyses with <1000 genes may lack resolution to distinguish short branches from multiple nodes[47,78]. For the plastome dataset, since all coding and non-coding regions are treated as a single heritable unit, no separate gene tree was inferred. We used IQ-TREE to build a ML tree for plastome dataset, and only the site consistency factor (sCF) was calculated.

### Phylogenetic network estimation

To explore whether gene flow contributed to cytonuclear discordance in *Artemisia*, we inferred and simulated phylogenetic networks using the maximum pseudo likelihood approach, accounting for incomplete lineage sorting (ILS) and gene flow. Phylogenetic networks were reconstructed from gene trees using the InferNetwork-MPL command in PhyloNet v.3.6.9[48]. The cytonuclear discordance patterns (Fig. 3a) suggested hybridization events among *Artemisia* subgenera. Given the computational limitation of PhyloNet, we constructed a 22-species dataset, comprising 20 *Artemisia* species (representing all eight subgenera) and two outgroups (*Artemisiella*+*Chrysanthemum/Elachanthemum*; Fig. 3c and d). To enhance taxon representativeness, two independent 22-species datasets were generated for PhyloNet analysis. For each dataset, we performed 5 network searches allowing 0 to 6 reticulations, with 10 replicates per search. To determine the optimal reticulation number, we used InferNetwork-MP to calculate the minimum depth of traceability (MDC) for different reticulation counts. The optimal model was identified as the reticulation number corresponding to the lowest MDC value and steepest decline in the MDC curve[79].

### Evolutionary analysis of macro- and micro-morphological characters

We investigated a total of 20 morphological characters potentially useful for *Artemisia* infrageneric taxonomy, of which 13 are macro-morphological (Supplementary Table 4) and seven are micro-morphological (Supplementary Table 5). For macromorphological characters, in addition to these previously used for subgeneric taxonomy[31], such as life form, capitulum type, synflorescence, leaf shape, and leaf size, we also investigated others, including plant height, capitulum diameter, leaf area, leaf length-width ratio, leaf segment length and leaf segment width (Supplementary Table 4, Supplementary Data 3). We introduced a character "leaf types" defined by the following seven specific leaf morphological features: leaf shape, number of leaf segment pairs, leaf size, leaf area, leaf length-width ratio, leaf segment length, and leaf segment width (Fig. 4 and Supplementary Fig. 12, Supplementary Table 3). We gathered data of these macromorphological characters for all the sampled 394 *Artemisia* and 6 outgroup species in the GM-derived ML tree (Fig. 2) through field observations and specimen examinations. Each character was measured in three individuals using ImageJ software (https://imagej.nih.gov/ij/), and the average value was calculated. Data were verified against literature (e.g., floras and protologues; Supplementary Data 1). For leaf size, a character reflecting overall leaf dimensions, we followed Cain[80] to derive a rough estimate: leaf size was approximated by measuring length and width, then calculating length × width × 3/4. In contrast, leaf area was defined as the precise one-side or projected area of an individual leaf[81]. Of the 13 macromorphological characters, eight are quantitative (plant height, capitulum diameter, number of leaf segment pairs, leaf size, leaf area, leaf length-width ratio, leaf segment length, and leaf segment width; Supplementary Data 4). To facilitate subsequent classification of character states, we discretized these eight continuous characters using a Gaussian Mixture Model (GMM)[82]. The optimal number of discrete states was determined by minimizing the Bayesian Information Criterion (BIC) in conjunction with the distribution of character states on the phylogenetic tree. The scripts used in this analysis are available on Figshare (https://doi.org/10.6084/m9.figshare.28164335). Ancestral states reconstruction for the 13

characters was performed using the maximum likelihood method in the "APE" package[83] of RASP v4.2[84] based on the GM-derived ML tree (Fig. 2). Phylogenetic signals of these characters were tested using Blomberg's $K$[85] and Pagel's $\lambda$[86] in R v3.6.1[87]. Phylogenetic signal denotes the tendency for closely related species to exhibit greater phenotypic resemblance than expected under random sampling from a phylogenetic tree. A strong signal is indicated when Pagel's $\lambda$ approaches 1 or Blomberg's $K$ exceeds 1.

We studied seven micromorphological characters of florets across all sampled *Artemisia* species and selected outgroup species (Supplementary Table 5), using our field collections and herbarium specimens deposited in PE, LE and MW. Sampling encompassed all eight subgenera and 24 sections (Supplementary Data 2 and 5), with three individuals examined per species. Capitula were processed through: (1) 24-hour FAA fixation; (2) ultrasonic cleaning (100 Hz, 5 min); (3) 5% NaOH treatment (2 hr). After rinsing, capitula were mounted in Hoyer's solution and imaged using a Leica DM5000B microscope. We examined the corolla and style of both marginal and disk florets, and the anther of disk florets. Terminology follows Roque et al[88]. and Grossi et al[89]. Due to material limitations and experimental constraints, micromorphological data were unavailable for some species in the GM-derived ML tree (Fig. 2). These species were excluded from the GM-based ML tree. Based on this pruned tree [200 *Artemisia* species and 4 outgroup species (*Chrysanthemum* + *Ajania*); Supplementary Data 5], ancestral states and phylogenetic signals of the seven micromorphological characters were reconstructed and estimated using the methods described above.

### Compilation of *Artemisia* species with taxonomic positions

We collected all accepted species names of *Artemisia* primarily from major databases: Plants of the World Online (https://powo.science.kew.org/); Global Compositae Checklist (https://www.compositae.org); World Flora Online (https://www.worldfloraonline.org); Catalog of Life (https://www.catalogueoflife.org); all retrieved on 29 September 2024. These names were cross-referenced with floras of its main distribution ranges (Supplementary Data 1) and our original research on herbarium specimens. We checked and standardized the species names following the *International Code of Nomenclature for algae, fungi, and plants*[90]. Based on the phylogenetic and morphological evidence (Fig. 2, Supplementary Figs. 12 and 13), we assigned each species to a subgenus and section within the proposed taxonomical framework (see Supplementary Data 6). Species with insufficient morphological and molecular evidence were treated as unplaced and listed separately. For conflicting taxonomic treatments, decisions were made based on our morphological assessments in the field and herbaria, along with the latest phylogeny established here (Fig. 2).

### Reporting summary

Further information on research design is available in the Nature Portfolio Reporting Summary linked to this article.

## Data availability

All data generated or analysed in this study are included in this published article and/or its supplementary materials. Sequencing data generated in this study have been deposited in GenBank under Bioproject PRJNA909040. All sequences used in the phylogenetic analyses are available in GenBank, with accession numbers listed in Supplementary Data 2. Morphological character data generated in this study is provided in Supplementary Data 3–5. The sectional taxonomy of *Artemisia* with an identification key to all the sections is provided in Supplementary Note 1. The accepted species list of *Artemisia* with subgeneric, sectional positions (including supporting evidence), and comparisons with databases and previous taxonomies are provided in Supplementary Data 6. Phylogenies and datasets generated during

and/or analysed in this study have been deposited in figshare data repository (https://doi.org/10.6084/m9.figshare.28164335).

## Code availability

We used publicly available software packages and scripts as described in the materials and methods. The scripts used for discretization of continuous morphological character are available on Figshare (https://doi.org/10.6084/m9.figshare.28164335).

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

## Acknowledgements

We thank the curators of A, B, BC, BCN, BR, E, GH, HAW, K, KUN, KYO, LE, MHA, MO, MW, P, PE, PTBG, TI and US, who granted us access to their collections. We also thank Dr. Zheng Li from Miami University for his assistance in polishing the language. This research was partly supported by the National Natural Science Foundation of China (Grant No. 32270229, 31870179 and 31570204 of T.G.G., 32361133549 of W.W.), Sino-Africa Joint Research Center (Grant No. SAJC201614 of T.G.G.), National Plant Specimen Resource Center (Grant No. E0117G1001 of W.W.), Tencent Foundation and Shan Shui Conservation Center of M.W.

## Author contributions

T.G.G. and B.H.J. conceived the study; B.H.J., T.G.G., M.W., G.H.N., Y.F.L., C.C. and J.Y.Z. collected and analyzed the samples; M.W., G.Y.H., J.H.S., J.Y.Z. and G.H.N. provided pictures; B.H.J. and X.Y.C. drew the figures and tables; T.G.G., J.V., F.J., D.V., F.V., A.S.E., A.P.S., A.S., A.L., X.J.G., G.K., and W.W. contributed botanical knowledge; B.H.J. and T.G.G. wrote the paper with inputs from all the authors.

## Competing interests

The authors declare no competing interests.
