## [Peer Review file · Nature Communications]

Global phylogeny and taxonomy of *Artemisia*

Corresponding Author: Dr Tiangang Gao

Version 0:

Reviewer comments:

Reviewer #1

(Remarks to the Author)

The manuscript describes what the authors consider the most comprehensive global phylogeny to date of the large genus *Artemisia* and its closest allied genera. The authors generate multiple genomic datasets encompassing low-copy nuclear genes, complete plastomes, and nuclear ribosomal ITS/ETS markers from different groups of sampled species and then combine those data into a gigamatrix comprising concatenated data. These datasets are subjected to different phylogenetic analyses and statistical tests to explore phylogenetic relationships, taxonomic placements, and hypothesized reticulations or hybridizations in the evolution of the genus. In addition, the authors also reappraise a suite of macro- and micromorphological characters for their taxonomic utility and evolutionary significance. The manuscript follows from a previous phylogenomic and morphological study of the genus (Jiao et al. 2023) by many of the same authors.

I consider this study a valuable contribution to advancing the understanding of this large, cosmopolitan, and complex genus and other large and taxonomically complex groups of organisms. However, I have some specific comments, which are provided below.

Specific comments:

The portrait layout of the supplementary tables makes reading and reviewing cumbersome; some columns and fields are cut off.

-throughout the manuscript, the term “palmar” describes leaf types. Shouldn't this be “palmate”?

-line 202: “Pecticae” should be Pectinatae

-line 277; Figure 4: measurement is 50 cm for plant height, but the cutoff in ST3 and other places is 30 cm. Please verify for consistency

-Some redundancy in the Discussion regarding statements about taxonomy and the proposed framework

-The overall methodology is sound, but I admit not being familiar with certain scripts and tests that specific software packages apply. However, the workflow steps from sampling to validation and analysis were logical to me. Some additional details could be provided to help with the interpretation of the findings.

-Concerning sampling methods and Table S2, the authors state that a single individual represents each species. Was this only for molecular sampling? What might the implications be for species with wide variation and divergent interpretations, such as *A. vulgaris*? The authors state that POWO and GCC provided guidance, but even these checklists do not have errors in what names are accepted versus synonyms. In Table S2, it would be helpful to indicate which specimens were newly collected from field investigations versus historical herb specimens. It's unclear to me based on the table's formatting, making the information hard to read.

-For the study of macro and micromorphology and their accompanying figures and tables, it was unclear whether the data were collected from single individuals or multiple plants or sources for molecular sampling. Are the morphological data from a single measurement, or are they averages from multiple measures per sample(s)? Some additional details on how these data were collected are needed. I assume data in Tables S4 & S5 were not all collected from the same individual, but some

from the field, some from available herbarium specimens, and some from literature, correct?

-line 527: Supplementary Table S2, not S1, provides sample information

-line 539-40: How was diploid determined? I didn't see the diploid species indicated in Table S2 as stated.

-lines 594: Confirming species identification by sequence alignments and excluding those with "highly unlikely phylogenetic positions, such as a different subgenus." Were any species excluded, and if so, how many? The logic of this is unclear, as are the potential implications given that conflicting subgeneric placements are not uncommon in *Artemisia*, whether looking at historical or contemporary treatments. This manuscript's revised framework places many species in entirely new subgeneric positions.

-The section beginning on line 628 says that 17 species were chosen as representatives for subgenera. How were these species selected?

-Lastly, the authors acknowledge that their updated infrageneric taxonomy (SI) is provisional, which I agree with. However, I would prefer to see this published separately to make it more accessible to those working on taxonomy and nomenclature. Presently, this treatment is lost as a supplemental file in this paper, with its emphasis on all the other phylogenetic data. It's just my opinion, and my decision to accept the manuscript doesn't hinge on this, but I feel the taxonomic treatment is a significant contribution and hypothesis for future work and could stand alone.

Reviewer #2

(Remarks to the Author)

The manuscript about the phylogeny and taxonomy of *Artemisia* is a major step forward in the knowledge of this large and important genus. The gigamatrix approach of integrating much new data using one of the currently best available data types with older molecular data of smaller analyses (and these were often with a special geographic focus) yielded exhaustive and very well supported and very reasonable phylogenetic hypotheses for the whole genus. The authors did not stop here but also included macro- and micromorphological data from the literature and their own observations. I am not familiar with all of the analyses performed but from my own knowledge and the explanations given by the authors the analyses were reasonable and correctly applied. The resulting partly new and now very well supported circumscription of the genus and the subgenera and sections is very important and opens several avenues for more research in *Artemisia*. I have few notes, however, that should be considered by the authors.

The manuscript is in my opinion generally well prepared and the figures are informative. I slightly wonder, however, why none of the many authors have noted the obvious and clear mistakes in the legends. These are:

Figure 1. There is a dot in South Africa but the country is white and not in a shade of green. How is that possible? The number of dots is clearly lower than 346. I guess many of which are on top of each other, e.g., in East Asia?

Figure 2. I am pretty sure that this is not NR data but NLC data (in figure and legend).

Figure 3 legend. Some photos do not show the habitat but the habitus.

Figure 4 capitula arrangement is not 1. Panicle; 2. Raceme; 3. Corymb, but rather 1. Corymb; 2. Panicle; 3. Raceme.

Figure 4 Also capitula type 2. Absinthium type needs to be replaced by 3. Dracunculus type and vice versa.

I have shortly checked these problems in the data and other parts of the figures and it seems that these were just mere slips of the pen in the figures and not in the data, but the authors might re-check in more detail whether this is true.

In the legend of Figure 1 explicitly useful species were described. One could a word in brackets in which respect each species is useful? References for the Global compositae checklist and the World Geographic Scheme should be given in the legend.

Line 108 give one example in one sentence.

Line 223 I might have overlooked it but the *Artemisiella* discordance and placement problem was not picked up in the discussion? As this seems the only problem remaining for genus circumscription this needs to be done.

The authors state that hybridization is partly responsible for the incongruencies found between data sets. Interestingly, some events happened in stem groups only but their tests and discussions even find evidence for hybridization in morphological characters of existing lineages. In conclusion, the overall treatment of hybridization, a problem that troubled previous work, was convincing for me. What I slightly missed was whether remnants of the specific hybridization signals found between cp and nuclear data could also be found within the nuclear data alone? The authors might discuss the gCF values for that in more detail?

Line 598 There were obviously some problems with contamination and misidentification. It should be stated explicitly how many samples were affected and in which cases re-identification was possible.

As stated before this analysis of *Artemisia* is well prepared and very valuable for future work on the very many species of the genus, the systematics of related genera and for evolutionary phenomena in general. However, in my view the authors slightly overdue the actual meaning of their work in their wording. You could delete lines 47 to 56 and 77-79 without lowering the actual impact. Also Figure 6 and some of the discussion connected to that figure is not strictly necessary in my

view and could be deleted.

Reviewer #3

(Remarks to the Author)

The manuscript "A global phylogeny and taxonomy of *Artemisia*: new insights from genome and morphology" by Jiao et al. presents the largest phylogeny of the genus *Artemisia* to date by combining and re-analysing existing sequence data in a "gigamatrix" approach, comprising just above two thirds (68.7%) of the 504 accepted species. For the gigamatrix, existing genome skimming data of 228 species that were generated by the same authorship (<https://doi.org/10.1093/aob/mcad051>) are re-analysed to retrieve plastomes and low-copy nuclear genes, and combined with the nuclear ribosomal markers ITS and ETS from published sanger sequence data. This gigamatrix phylogeny is combined with a large and mostly newly generated morphological dataset comprising 12 macromorphological characters for 366 species, including at least five new characters, and seven micromorphological characters for 200 species to produce a new classification of the genus into eight subgenera and 24 sections. The methodology is described in detail and overall easy to follow.

The paper is generally well-written and centered around the challenging problem of bringing taxonomic structure into species-rich taxa such as *Artemisia*. In my view, the most important achievement of the paper is generating a novel global classification that places 501 (99.4%) of all 504 accepted species within a section of the genus, breaking down this diverse clade into more easily manageable groups that can be studied individually, and thus facilitating future targeted research of individual subunits within *Artemisia*. I would like to ask the authors to address the following main points:

- 1) I think that the balance of the results presented in the paper is not quite right: the phylogeny is presented as a major achievement (e.g. "1. The most comprehensive molecular phylogenetic sampling", "3. The most comprehensive phylogeny of *Artemisia* to date") but appears to be almost entirely based on existing sequence data from previous studies and should thus be presented more modestly in my opinion. In contrast, the morphological dataset appears to be largely new, and this important achievement could receive more emphasis.
- 2) I also think that more work needs to be done to contextualise the new phylogeny and classification in relation to existing studies. For example, which previously inferred relationships and classifications are confirmed, and which ones are contradicted or newly resolved?
- 3) As a technical question, I wonder why a concatenated tree was chosen to present phylogenetic relationships in Figs. 2- 4 instead of a multi-species coalescent (ASTRAL) tree that accounts for potentially differing evolutionary trajectories of individual genes? The methods state that both a concatenated and an ASTRAL species tree were built, but I think at least an explanation or justifications needs to be added as to why one tree is presented instead of the other.
- 4) Finally, I think it is debatable whether the approach of combining phylogeny and morphology to create a classification is truly novel as claimed as similar approaches have been taken before (e.g., see Muñoz-Rodríguez et al. 2023, <https://doi.org/10.1002/tax.12887>) and suggest that any such claims should be toned down or removed.

I also attach the main and supplementary files with additional minor points for consideration by the authors.

I hope that these comments will be useful for further improving the quality of the paper and to present these significant achievements in the most appropriate way.

Dr Benedikt Kuhnhäuser
Royal Botanic Gardens, Kew

Reviewer #4

(Remarks to the Author)

This study represents an updated addition to the understanding of the evolutionary relationships and classification of the genus *Artemisia*. It takes on a previous study by Jiao and collaborators from 2023, in which they created a large genome skimming dataset to study the evolutionary relationships in this mega diverse genus. The dataset is recycled in this new manuscript, but data exploitation is based on an alternative approach to use genome skimming data, extracting low copy genes, chloroplast data and nrDNA loci.

Overall, the manuscript is well written and cohesive, the results are strong and thoroughly discussed, but I have some comments for the authors to consider.

-In my opinion, this is an interesting approach, but I somehow miss throughout the text a justification about why following this new approach is essential for understanding *Artemisia*, if the results from this study mirror to a large extent those from Jiao et al 2023, with most of the taxonomical implications on the new classification following those from the study from 2023. I believe adding an statement emphasizing on this point will strengthen the relevance and impact of this new study, especially for a multidisciplinary journal such as *Nat. Comms*.

-I am also wondering why the SNPs information from Jiao et al. (2023) has not been used in this megamatrix approach, if at all possible, which might not be the case, but coding this information into the matrix could also provide additional strength to the analysis. The study provides, though, a significant increase of additional morphological traits from that previous study, which I have no doubts will be very useful to inform future taxonomical studies.

-I understand that sampling from highly diverse genera such as *Artemisia* is challenging and can be proved difficult. Certainly, the expanded sampling of 138 species in this new study refers mostly (if not all) to Genbank data on ITS/ETS nrDNA sequences, and to some extent means that all this additional phylogenetic information relies on the analysis of two

loci. Could this imply any sort of analytical bias when combined with the NLC dataset? One has to bear in mind that for multiple species the analysis relies on c. 900 bp whilst in other sequence information spans over 216 loci (c. 260000 bp). I would suggest the authors to comment on this. The phylogenetic approaches used in this manuscript are beyond my field of expertise, so I am not at all criticizing such take on, but consider a discussion on this point important to the readers to better understand any technical flaws arising from such strategies.

-I was surprised to find no reference to data availability along the manuscript regarding the resulting trees and associated metadata, which I believe should be also made available if the ultimate aim is for the scientific community to use in the future the resources generated in *Artemisia*. Based on that, the authors should add a statement as to whether editable versions of the megamatrix datasets and trees are, or will be, deposited in open repositories.

-The authors claimed that besides the NLC dataset (Low copy genes from genome skimming), they also built a NR dataset (ITS+ETS extracted from genome skimming), plus an additional dataset with ITS+ETS data from Genbank (Table S2). Fig. S1 (concatenated ML tree of NR data) only includes data from the NR dataset (i.e. genome skimming), and I understand that Fig. S9. (concatenated ML tree of the NR Data) includes data from the NR dataset and Genbank, however, these two files have the same title, yet the Fig. S9 is not cited anywhere in the main manuscript (unless I have missed it). This creates some sort of confusion to the reader. For example, in page 9 line 197-99 reads "2) In the plastome trees, seven out of the eight subgenera of *Artemisia* recognized in the NLC tree are not monophyletic, except for *A. subg. Pacifica*, for which a single species was sampled (Fig. 2a)." I understand that in the NLC only one species of subg. *Pacifica* (*A. chinensis*) has been sequenced, but ITS and ETS data is available for the other 3 species in the subgenus, as shown in Fig. 3 and Fig. S9, therefore the monophyly is confirmed with more than one species. This point could be further explained in the manuscript, and perhaps merge Fig. S1 and S9 into one, since both represent ITS/ETS data regardless of their origin (genbank vs. skimming)

-In page 17, line 400-402, "For instance, *A. magellanica*, a newly sampled Patagonian species previously assigned to *A. subg. Artemisia* (ref. 77), was revealed to belong to *A. subg. Pectinatae* (Fig. 3)" After careful reading of the cited study (Ref. 77, Garcia et al. 2011) I am convinced that it represents a secondary reference, since the sequence data on *A. magellanica* used comes from an original study on focused on South American *Artemisia* by Pellicer et al. (2010), and therefore the following citation should be added instead of Ref. 77. (Pellicer, J., T. Garnatje, J. Molero, F. Pustahija, S. Siljak-Yakovlev, and J. Vallès. 2010b. Origin and evolution of the South American endemic *Artemisia* species (Asteraceae): Evidence from molecular phylogeny, ribosomal DNA and genome size data. *Australian Journal of Botany* 58: 605–616.)

Version 1:

Reviewer comments:

Reviewer #1

(Remarks to the Author)

Upon review of the revised manuscript and rebuttal, I am satisfied with the changes and improvements made by the authors in response to my comments. The authors provided detailed rebuttals to each of my questions and made appropriate corrections to the text, figures, and tables as suggested, including supplemental files. As stated in my initial review, I consider this study a valuable contribution to advancing the understanding of this large, cosmopolitan, and complex genus, and I see value in applying the approach to other large and taxonomically-complex groups of organisms. The resulting global phylogeny is the most comprehensive to date for *Artemisia* and its allied genera. I also agree with the authors in that their taxonomic classification, although provisional as classifications tend to be, is insightful and will serve as a helpful framework for more detailed studies at the species level going forward.

Nevertheless, I found one minor correction that should be made regarding the color codes in Figure S6. The color code for *Artemisia subg. Pacifica* is "orange" in the legend and elsewhere in the paper, but the lines in the phylogeny for these taxa are black in Fig. S6. I made this observation while looking over comments from Reviewer #4 in the author's rebuttal. Otherwise, I recommend acceptance of the revised manuscript.

(Remarks on code availability)

I looked over the files using the provided link, but I did not try to install or run any programs. I also went through the list of codes and opened them to check for access, and I didn't find any noticeable issues. The data, codes, phylogenies, etc all appeared to be available and accessible to anyone who would want to use them.

Reviewer #2

(Remarks to the Author)

The manuscript has been carefully revised and I am content with the changes made or the explanations given by the authors when they were of different opinion. In my view it is fit for publication now.

(Remarks on code availability)

Reviewer #3

(Remarks to the Author)

The authors have addressed all of my comments with great attention to detail and have made several substantial changes to the manuscript in response. I have no further suggestions and congratulate the authors to their work!

Dr Benedikt Kuhnhäuser
Royal Botanic Gardens, Kew

(Remarks on code availability)

I tried to access the provided link under "Code availability" (<https://doi.org/10.6084/m9.figshare.28164335>) but was directed to a page displaying "Error: DOI not found". Maybe the DOI is not active yet? As such, I have not been able to look at the code.

Beyond this, however, the paper largely uses publicly available software and describes the methodology in sufficient detail, so that I am satisfied that the methodology is generally transparent and reproducible.

Response to Reviewers

Reviewer #1 (Remarks to the Author):

The manuscript describes what the authors consider the most comprehensive global phylogeny to date of the large genus *Artemisia* and its closest allied genera. The authors generate multiple genomic datasets encompassing low-copy nuclear genes, complete plastomes, and nuclear ribosomal ITS/ETS markers from different groups of sampled species and then combine those data into a gigamatrix comprising concatenated data. These datasets are subjected to different phylogenetic analyses and statistical tests to explore phylogenetic relationships, taxonomic placements, and hypothesized reticulations or hybridizations in the evolution of the genus. In addition, the authors also reappraise a suite of macro- and micromorphological characters for their taxonomic utility and evolutionary significance. The manuscript follows from a previous phylogenomic and morphological study of the genus (Jiao et al. 2023) by many of the same authors.

I consider this study a valuable contribution to advancing the understanding of this large, cosmopolitan, and complex genus and other large and taxonomically complex groups of organisms. However, I have some specific comments, which are provided below.

Response 1: Thank you very much for your positive comments, especially you highlight that "this study a valuable contribution to advancing the understanding of this large, cosmopolitan, and complex genus and other large and taxonomically complex groups of organisms."

The portrait layout of the supplementary tables makes reading and reviewing cumbersome; some columns and fields are cut off.

Response 2: Thank you for highlighting this issue. We have converted the large supplementary tables into Supplementary Data files to comply with *Nature Communications'* formatting requirements. This adjustment should prevent truncation and improve readability.

-throughout the manuscript, the term "palmar" describes leaf types. Shouldn't this be "palmate"?

Response 3: We agree and have thoroughly reviewed the manuscript and corrected all instances of "palmar" to "palmate" throughout the text (Tables S3 and S4 in Supplementary Information).

-line 202: "Pecticae" should be Pectinatae

Response 4: Changed as you suggested (Line 246).

-line 277; Figure 4: measurement is 50 cm for plant height, but the cutoff in ST3 and other places is 30 cm. Please verify for consistency

Response 5: Sorry for this inconsistency. Based on the suggestion of Reviewer 3, we have refined our approach and re-standardized the cutoff for plant height from 30 cm to 50 cm in the revised version (Table S4, Lines 558–564; see more details in Responses 2 to Reviewer 3 below).

-Some redundancy in the Discussion regarding statements about taxonomy and the proposed framework

Response 6: We appreciate the feedback. As recommended, we have removed the overlapping content regarding taxonomy and the proposed framework. The revised Results and discussion section now exclusively retains taxonomy-related content under the subheading "The comprehensive sectional taxonomy of *Artemisia*" (Lines 321–352).

-The overall methodology is sound, but I admit not being familiar with certain scripts and tests that specific software packages apply. However, the workflow steps from sampling to validation and analysis were logical to me. Some additional details could be provided to help with the interpretation of the findings.

Response 7: In the revised manuscript, we enhanced the Methods section to improve clarity, specifically by:

1) providing software version numbers and script repositories for all custom analyses (accessible at doi: <https://doi.org/10.6084/m9.figshare.28164335>);

2) detailing the taxon selection criteria in *PhyloNet* (under "Phylogenetic network estimation", Lines 531–534);

3) specifying sample sizes and standardizing morphological measurements protocols (Lines 552–555);

4) elaborating on the rationale and methodology for converting quantitative to qualitative characters (Lines 558–564).

All data generated or analyzed in the course of this study have been included in the revised manuscript and supplementary materials, with a dedicated Data availability section and Code availability section added (Lines 598–613). Very good suggestion, thank you!

-Concerning sampling methods and Table S2, the authors state that a single individual represents each species. Was this only for molecular sampling? What might the implications be for species with wide variation and divergent interpretations, such as *A. vulgaris*?

Response 8: Thank you for your insightful query regarding our sampling strategy. In our molecular sampling, a single-individual-per-species approach was adopted to establish a global infrageneric framework for *Artemisia*—a foundational step preceding fine-scale species delimitations. This focus on species-level sampling was prioritized to resolve broad phylogenetic relationships and reconstruct infrageneric taxonomy, rather than addressing within-species variation.

For notoriously variable species complexes like *A. vulgaris*, our current sampling does not resolve its species delimitations, which may lead to shifts in recognized species circumscriptions. This consideration applies to other challenging *Artemisia* complexes (e.g., *A. tridentata* complex, *A. maritima* complex). Therefore, we stated at the end of Results and discussion section: "Notably, our single-representative-per-species approach does not address species delimitations in *Artemisia*, and thus this species treatment herein remains provisional, consistent with the inherently dynamic nature of taxonomic studies." (Lines 349–352).

Species delimitation in such complexes requires intensive within-species sampling, a resource-intensive endeavor. For the *A. vulgaris* complex, we are actively addressing this through ongoing fieldwork across diverse regions (UK, Belgium, Czech Republic, Russia, China, and the United States), with a large collection of geographically and morphologically representative samples. Future plans involve integrating expanded genomic and morphological datasets to refine species boundaries in this and other variable lineages. The taxonomic framework established here serves as a pivotal foundation for targeted species-level studies in these groups, as discussed in the Conclusions section (Lines 367–368).

The authors state that POWO and GCC provided guidance, but even these checklists do not have errors in what names are accepted versus synonyms. In Table S2, it would be helpful to indicate which specimens were newly collected from field investigations versus historical herb specimens. It's unclear to me based on the table's formatting, making the information hard to read.

Response 9: We regret the formatting inconsistencies in Table S2, which has been reformatted as Supplementary Data 2 to enhance readability. Per suggestion, we added a "Material type" column distinguishing field-collected and herbarium specimens. Taxonomic discrepancies with major databases (POWO, GCC, WFO, COL) are now analyzed in Supplementary Data 6. Potential inaccuracies in databases (e.g., POWO, GCC) were addressed by cross-referencing recent floras (see Supplementary Data 1) and verifying herbarium specimens by our team. Recognizing that taxonomy evolves with new evidence, all data in this manuscript reflect current knowledge as of December 2024 (Lines 585–597).

-For the study of macro and micromorphology and their accompanying figures and tables, it was unclear whether the data were collected from single individuals or multiple plants or sources for molecular sampling. Are the morphological data from a

single measurement, or are they averages from multiple measures per sample(s)? Some additional details on how these data were collected are needed. I assume data in Tables S4 & S5 were not all collected from the same individual, but some from the field, some from available herbarium specimens, and some from literature, correct?

Response 10: Yes, as you correctly noted, the data in Tables S4 and S5 (renumbered as Supplementary Data 4, 5 in the revised manuscript) are based on measurements from multiple individuals, including field collections and herbarium specimens. For each morphological character, measurements were taken from at least three individuals where feasible. Data were validated against floras, protologues, and documented records. For narrowly distributed species with limited specimens (1–2 available), measurements involved fewer than three individuals. These methodological details are now documented in the Methods section (Lines 565–569).

-line 527: Supplementary Table S2, not S1, provides sample information

Response 11: Changed as you suggested (now renumbered as Supplementary Data 2, Lines 127 and 376).

-line 539-40: How was diploid determined? I didn't see the diploid species indicated in Table S2 as stated.

Response 12: We are glad for this opportunity to clarify this issue. We determined diploid species via cytological studies. In the revised version, we added the methodology in the Supplementary Methods of Supplementary Information as follows:

Plants were first cultivated in the greenhouse of the Institute of Botany, Chinese Academy of Sciences, Beijing. Fast-growing root tips were then cut, pretreated in a solution with 8-hydroxyquinoline and colchicine in a ratio of 1:1 for 3 hours in the dark, and fixed in Carnoy I (glacial acetic acid: absolute ethanol, 1:3) at 4 °C for 1 h. Next, they were macerated in 1 mol/L hydrochloric acid at 65°C for 8 minutes, and then soaked in clean water for more than 10 minutes, and squashed for chromosome observations. Chromosome counts were performed using a Leica DM1000 microscope (Leica Microsystems, Wetzlar, Germany), with images documented in Supplementary Information (Fig. S16) of the revised manuscript (Line 424). All identified diploid species determined by us are marked with an asterisk (*) before their names in Supplementary Table 2 (now renumbered as Supplementary Data 2 in the revised manuscript). Chromosome counts for these taxa were cross-validated against the Chromosome Counts Database (CCDB, Rice et al., 2015 *New Phytol.* 206, 19–26).

-lines 594: Confirming species identification by sequence alignments and excluding those with "highly unlikely phylogenetic positions, such as a different subgenus." Were any species excluded, and if so, how many? The logic of this is unclear, as are the potential implications given that conflicting subgeneric placements are not

uncommon in *Artemisia*, whether looking at historical or contemporary treatments. This manuscript's revised framework places many species in entirely new subgeneric positions.

Response 13: Thank you for your meticulous review and thoughtful query. In fact, only one sample (originally labeled as *Artemisia palmeri*, re-identified as *A. absinthium*; voucher: T. G. Gao 7613, PE) was excluded due to highly unlikely phylogenetic position. Our initial genomic data placed it in *A. subg. Absinthium*, contradicting morphological evidence (Shultz, 2006 *Flora of North America* 503–534) and prior phylogenies (Garcia et al., 2011 *Am. J. Bot.* 98, 638–653; Hobbs & Baldwin, 2013 *J. Biogeogr.* 40, 442–454), which uniformly support its position in *A. subg. Artemisia*. This discrepancy suggested potential mislabeling, prompting its intended exclusion. However, an oversight retained it in the original ML tree (Fig. 3 in our initial submission). To resolve this, we incorporated GenBank sequences of *A. palmeri* (Accession numbers: HQ019052, JX051740), which confirmed its placement in *A. subg. Artemisia* (Fig. 2 in the revised manuscript). Quality control of other samples from the same batch identified no similar issues. The mislabeled sample was removed in the revised manuscript to adhere to our stated strategy of one sample sequenced per species. As this issue is now resolved, we omitted the related discussion to streamline the manuscript. We hope that you will find these revisions satisfactory.

-The section beginning on line 628 says that 17 species were chosen as representatives for subgenera. How were these species selected?

Response 14: We appreciate this critical inquiry. In the revised manuscript, we expanded the NLC dataset by adding genome skimming data from 96 additional *Artemisia* species, increasing PhyloNet taxon sampling from 17 to 22 species. Given the cytonuclear discordance indicating inter-subgeneric hybridization, we randomly selected 20 species to represent all eight subgenera of *Artemisia* (1–3 species per subgenera; Fig. 3) and two outgroup taxa, forming a 22-species dataset (20 ingroups + 2 outgroups) for PhyloNet analysis. These details—including species selection to maximize subgeneric coverage while addressing computational constraints—are now described in the Methods section (under "Phylogenetic network estimation", Lines 531–534).

-Lastly, the authors acknowledge that their updated infrageneric taxonomy (SI) is provisional, which I agree with. However, I would prefer to see this published separately to make it more accessible to those working on taxonomy and nomenclature. Presently, this treatment is lost as a supplemental file in this paper, with its emphasis on all the other phylogenetic data. It's just my opinion, and my decision to accept the manuscript doesn't hinge on this, but I feel the taxonomic treatment is a significant contribution and hypothesis for future work and could stand alone.

Response 15: We are grateful to the reviewer for emphasizing the critical role of taxonomic treatment. We fully agree that taxonomy serves as the foundation for biodiversity research, bioprospecting, conservation, and other related fields, as robust taxonomic frameworks are indispensable to these efforts. Due to the strict length constraints of *Nature Communications*, we had to prioritize the phylogenetic analyses and taxonomic framework in the main text, presenting the detailed taxonomic treatment in a supplementary file. However, we regard the taxonomic treatment inside as an important contribution and plan to fully publish it as a standalone paper in *TAXON* or a similar journal (anticipated 20+ printed pages), making it more accessible to taxonomists and systematists. The publication in *NC* could serve as a foundational phylogenetic and taxonomic framework that will underpin and accelerate the ongoing local taxonomic revisions in the world, aligning with our goal to advance *Artemisia* research.

Reviewer #2 (Remarks to the Author):

The manuscript about the phylogeny and taxonomy of *Artemisia* is a major step forward in the knowledge of this large and important genus. The gigamatrix approach of integrating much new data using one of the currently best available data types with older molecular data of smaller analyses (and these were often with a special geographic focus) yielded exhaustive and very well supported and very reasonable phylogenetic hypotheses for the whole genus. The authors did not stop here but also included macro- and micromorphological data from the literature and their own observations. I am not familiar with all of the analyses performed but from my own knowledge and the explanations given by the authors the analyses were reasonable and correctly applied. The resulting partly new and now very well supported circumscription of the genus and the subgenera and sections is very important and opens several avenues for more research in *Artemisia*. I have few notes, however, that should be considered by the authors.

Response 1: We appreciate your positive assessment of our analytical approach and resulting taxonomy, which validates our decades-long effort to tackle the persistent challenges in this cosmopolitan and taxonomically complex genus.

The manuscript is in my opinion generally well prepared and the figures are informative. I slightly wonder, however, why none of the many authors have noted the obvious and clear mistakes in the legends. These are:

Figure 1. There is a dot in South Africa but the country is white and not in a shade of green. How is that possible? The number of dots is clearly lower than 346. I guess many of which are on top of each other, e.g., in East Asia?

Response 2: We appreciate the opportunity to address this concern. The points represent georeferenced samples, with some coordinates overlapping due to collections from the same or proximate locations—particularly in species-rich regions like Asia—resulting in fewer visible dots than the total sample size. The dot in South

Africa corresponds to *Artemisia afra*, and we appreciate your keen observation of the map's shading inconsistency, an unintended artifact from the mapping process. In the revised version, we re-examined global species richness data, corrected the South Africa shading, and redrew the figure for clarity. Sampling latitude/longitude is provided in Supplementary Data 2, and regional *Artemisia* diversity data are available at figshare data repository (<https://doi.org/10.6084/m9.figshare.28164335>).

Figure 2. I am pretty sure that this is not NR data but NLC data (in figure and legend).

Response 3: All figure labels and captions have been corrected throughout the manuscript to ensure consistency and accuracy (Lines 842–849). We appreciate your careful review.

Figure 3 legend. Some photos do not show the habitat but the habitus.

Response 4: Thank you for pointing that out. Floret morphology photos highlighting capitulum types—a key characters in *Artemisia*—now replace habitat photos for representative species across all 24 sections. In the revised Figure 2 (the original Figure 3), dashed lines connect these to corresponding species, ensuring clear visual association (Lines 838–841). We hope that you will find these revisions satisfactory.

Figure 4 capitula arrangement is not 1. Panicle; 2. Raceme; 3. Corymb, but rather 1. Corymb; 2. Panicle; 3. Raceme.

Response 5: We agree and have corrected the terminology for capitulum arrangement (now termed "synflorescence" in the revised manuscript, Lines 269 and 545) in Figure 4, as detailed in Supplementary Table S4.

I have shortly checked these problems in the data and other parts of the figures and it seems that these were just mere slips of the pen in the figures and not in the data, but the authors might re-check in more detail whether this is true.

Response 6: We have verified all data and figures, confirming the issues were limited to minor labeling errors. These have been corrected throughout the manuscript.

In the legend of Figure 1 explicitly useful species were described. One could a word in brackets in which respect each species is useful? References for the Global compositae checklist and the World Geographic Scheme should be given in the legend.

Response 7: We expanded the legend of Figure 1 to specify species uses (medicinal, ecological, ornamental) and added citations for the Global Compositae Checklist and World Geographic Scheme (Lines 832–835).

Line 108 give one example in one sentence.

Response 8: As suggested, we added: "the homogamous-discoïd capitulum was revealed to evolve independently based on the evidence from cytology and molecular phylogenies^{31,32}. Yet all species bearing this state were once classified into a single genus *Seriphidium* based solely on their similar capitulum morphology³³." (Lines 92–95).

Line 223 I might have overlooked it but the *Artemisiella* discordance and placement problem was not picked up in the discussion? As this seems the only problem remaining for genus circumscription this needs to be done.

Response 9: We are glad for this opportunity to extend the discussion to address *Artemisiella*'s taxonomic placement and phylogenetic discordance (Lines 171–181). Analyses show strong support (BS = 100%) for *Artemisia* sister to (*Artemisiella* + *Phaeostigma*) in ML trees (GM/NLC datasets; Figs. S3 c and e); weak support (LPP = 0.61) for *Artemisiella* nested within *Artemisia* in ASTRAL analysis (NLC datasets; Figs. S3b, S5); alternative topology tests (SH/KH/AU) do not reject inclusion within *Artemisia* (Table S2). Given that *Artemisiella* can be distinguished from *Artemisia* by morphological characters (e.g., leaves oblong, 3-pinnatisect, with 8–13 pairs of lateral lobes), we tentatively retain *Artemisiella* as a separate genus pending further evidence.

The authors state that hybridization is partly responsible for the incongruencies found between data sets. Interestingly, some events happened in stem groups only but their tests and discussions even find evidence for hybridization in morphological characters of existing lineages. In conclusion, the overall treatment of hybridization, a problem that troubled previous work, was convincing for me. What I slightly missed was whether remnants of the specific hybridization signals found between cp and nuclear data could also be found within the nuclear data alone? The authors might discuss the gCF values for that in more detail?

Response 10: Thank you for your positive comments. We confirm hybridization signals from cytonuclear discordance are detectable within nuclear data. Genealogical concordance analyses show exceptionally low gCF (< 5%) at discordant nodes (e.g., *A. subg. Dracunculus*, *A. subg. Ponticae*; full data Fig. S2), versus higher gCF (53.5%) at concordant nodes (*A. subg. Pacifica*). This pattern indicates residual hybridization signatures in nuclear genomes. The revised " Cytonuclear discordance and possible hybridization within *Artemisia*" section (Lines 206–225) now reflects these findings. We hope that you will find these revisions satisfactory.

Line 598 There were obviously some problems with contamination and misidentification. It should be stated explicitly how many samples were affected and in which cases re-identification was possible.

Response 11: Only one sample (originally labeled as *Artemisia palmeri*, re-identified as *A. absinthium*; voucher: T. G. Gao 7613, PE) was excluded due to conflicting phylogenetic placement. Initial genomic data placed it in *A. subg. Absinthium*,

conflicting with morphological evidence (Shultz, 2006 *Flora of North America* 503–534) and prior phylogenies (Garcia et al., 2011 *Am. J. Bot.* 98, 638–653; Hobbs & Baldwin, 2013 *J. Biogeogr.* 40, 442–454), which uniformly support its position in *A. subg. Artemisia*. This discrepancy suggested potential mislabeling, prompting exclusion, though an oversight retained it in the original ML tree (initial Fig. 3). To resolve this, we incorporated GenBank sequences of *A. palmeri* (Accession numbers: HQ019052, JX051740), confirming its placement in *A. subg. Artemisia* (Fig. 2 in the revised manuscript). Quality control of other samples from the same batch identified no similar issues. The mislabeled sample was removed in the revised manuscript to adhere to our one-sample-per-species strategy, and related discussion was omitted to streamline the manuscript. We hope that you will find these revisions satisfactory.

As stated before this analysis of *Artemisia* is well prepared and very valuable for future work on the very many species of the genus, the systematics of related genera and for evolutionary phenomena in general. However, in my view the authors slightly overdue the actual meaning of their work in their wording. You could delete lines 47 to 56 and 77-79 without lowering the actual impact. Also Figure 6 and some of the discussion connected to that figure is not strictly necessary in my view and could be deleted.

Response 12: We appreciate the acknowledgment and have removed overstated passages (original Lines 47–56, 77–79) and Figure 6 with related discussion to focus on core findings. Adjacent text was revised for narrative coherence.

Reviewer #3 (Remarks to the Author):

The manuscript "A global phylogeny and taxonomy of *Artemisia*: new insights from genome and morphology" by Jiao et al. presents the largest phylogeny of the genus *Artemisia* to date by combining and re-analysing existing sequence data in a "gigamatrix" approach, comprising just above two thirds (68.7%) of the 504 accepted species. For the gigamatrix, existing genome skimming data of 228 species that were generated by the same authorship (<https://doi.org/10.1093/aob/mcad051>) are re-analysed to retrieve plastomes and low-copy nuclear genes, and combined with the nuclear ribosomal markers ITS and ETS from published sanger sequence data. This gigamatrix phylogeny is combined with a large and mostly newly generated morphological dataset comprising 12 macromorphological characters for 366 species, including at least five new characters, and seven micromorphological characters for 200 species to produce a new classification of the genus into eight subgenera and 24 sections. The methodology is described in detail and overall easy to follow.

Response 1: Thank you very much for your positive comments!

The paper is generally well-written and centered around the challenging problem of bringing taxonomic structure into species-rich taxa such as *Artemisia*. In my view, the most important achievement of the paper is generating a novel global classification that places 501 (99.4%) of all 504 accepted species within a section of the genus,

breaking down this diverse clade into more easily manageable groups that can be studied individually, and thus facilitating future targeted research of individual subunits within *Artemisia*. I would like to ask the authors to address the following main points:

1) I think that the balance of the results presented in the paper is not quite right: the phylogeny is presented as a major achievement (e.g. "1. The most comprehensive molecular phylogenetic sampling", "3. The most comprehensive phylogeny of *Artemisia* to date") but appears to be almost entirely based on existing sequence data from previous studies and should thus be presented more modestly in my opinion. In contrast, the morphological dataset appears to be largely new, and this important achievement could receive more emphasis.

Response 2: We appreciate your insight regarding the balance between phylogenetic and morphological contributions. In the revised manuscript, we expanded the morphological analysis with detailed descriptions (Lines 541–583) and merged the original sections ("The most comprehensive molecular phylogenetic sampling" + "The most comprehensive phylogeny of *Artemisia*") into one "An expanded phylogenetic framework incorporating 78% *Artemisia* species" (Line 124) to reduce redundancy and to tone the statement down.

In the revised manuscript, our key advances in the phylogeny of *Artemisia* over previous studies are summarized as follows:

Generated genome-skimming data for additional 96 *Artemisia* species, increasing the species coverage of *Artemisia* from 41% (205 species; Jiao et al., 2023 *Ann. Bot.* 131, 867–883) to 78% (394 species; this study) (Line 145; Fig. 2).

Acquired 202 nuclear low-copy genes from 314 species (298 *Artemisia* + 16 allies), surpassing Jiao et al. (2023)'s SNP data for 215 species (205 *Artemisia* + 10 allies) (Line 131; Fig. S2).

Obtained ITS/ETS sequences for 414 species (394 *Artemisia* + 20 allied species, 80% newly generated), exceeding Malik et al. (2017 *Taxon* 66, 934–952)'s 266 species (263 *Artemisia* + 3 allied species) (Line 132 and 464–469; Fig. S4).

Reconstructed the first plastome phylogeny for 314 species (298 *Artemisia* + 16 allied species, 59% representation) (Lines 196–204; Figs. 3, S6).

We think these advancements justify the term "comprehensive" where appropriately qualified. We hope that you will find these explanations and revisions satisfactory.

2) I also think that more work needs to be done to contextualise the new phylogeny and classification in relation to existing studies. For example, which previously inferred relationships and classifications are confirmed, and which ones are contradicted or newly resolved?

Response 3: We appreciate your emphasis on contextualizing our phylogeny and taxonomy with existing studies. In the revised manuscript, we systematically compared our findings with prior work, focusing primarily on Jiao et al. (2023 *Ann. Bot.* 131, 867–883)'s SNP-based analysis—the only robust global phylogeny of *Artemisia* with > 200 species—and taxonomic framework (Ling, 1982 *Bull. Bot. Res.* 2, 1–60; Ling et al., 2011 *Flora of China*, Vol. 20 1151–1259; primarily based on gross morphology). A comparative summary appears in Figure S3 and Supplementary Data 6.

Key Comparisons and Revisions:

1. Confirmed relationships and taxonomy

Monophyly of expanded *Artemisia* including 9 small genera (e.g., *Ajaniopsis*, *Crossostephium*); 8 strongly supported clades (Figs. 2 and S3), corresponding to 8 subgenera (Figs. 2 and S3, Lines 152–153); 24 subclades corresponding to 24 sections (Figs. 2 and S3; Lines 147–149).

2. Revised relationships and taxonomy

Subclade level: 11 subclades (e.g., *Salsoloides*, *Dracunculus*) and their sections (e.g., *Artemisia* sect. *Salsoloides*, *A.* sect. *Dracunculus*) were revised (Fig. 2; Supplementary Data 3; Lines 151–152).

Species level: 21 species transferred from *A.* subg. *Artemisia* to *A.* subg. *Dracunculus* supported by GM phylogeny and shared morphological characters (Figs. 2, S1, S3, S4; Lines 159–170).

3. Newly resolved clades and new taxonomy

Ten newly identified subclades (e.g., *Auratae*, *Laciniatae*) and 10 new sections (e.g., *A.* sect. *Auratae*, *A.* sect. *Norvegica*) (Fig. 2; Supplementary Data 3; Lines 151–152); Inclusion of *Ajaniopsis* into *A.* subg. *Absinthium* sect. *Argyrophyllae* (Fig. 2; Supplementary Data 3; Lines 174–177).

4. Unresolved

Artemisiella placement (conflicting signals; retained as separate genus based on morphology) (Figs. S3, S5; Table S2; Lines 182–192).

5. Limitations

Current species sampling, genomic sequencing, and morphological analyses require expansion, rendering the present taxonomic framework provisional. Future studies with new evidence will test and revise this phylogeny and taxonomy (Lines 348–352).

3) As a technical question, I wonder why a concatenated tree was chosen to present phylogenetic relationships in Figs. 2–4 instead of a multi-species coalescent (ASTRAL) tree that accounts for potentially differing evolutionary trajectories of individual genes? The methods state that both a concatenated and an ASTRAL species tree were built, but I think at least an explanation or justifications needs to be added as to why one tree is presented instead of the other.

Response 4: We appreciate this critical technical inquiry. In the revised manuscript, we added a topological comparison of nuclear trees (ASTRAL and concatenated ML) in Fig. S3, with an explanation for presenting concatenated ML trees in the main text (Lines 141–147), summarized as follows:

Data heterogeneity constraints: The gigamatrix combines nuclear genomic data (abundant sequences, fewer species) and ribosomal DNA (more species, sparser sequences), creating heterogeneity that rendered multi-species coalescent analysis (ASTRAL) computationally infeasible and statistically unreliable for the full dataset (Portik et al., 2023 *Mol. Biol. Evol.* 40, msad109; Lines 496–498).

Topological robustness: While both concatenated and ASTRAL analyses were performed on NLC dataset (Figs. S2 and S5), concatenated trees exhibited more stable and robust support for major *Artemisia* clades compared to coalescent-based methods (mean 99% vs. 58% in ASTRAL, see Fig. S3 for comparison). Besides, the backbone of the ASTRAL tree is consistent with concatenated ML tree, differing in that the genus *Artemisiella* was embedded within *Artemisia*, and both *A. subg. Dracunculus* and *A. subg. Ponticae* were polyphyletic (Fig. S3), with discordant nodes having low local posterior probability (LPP < 0.5, Fig. S3), indicating minimal conflict. Given this congruence and the superior robustness of concatenated topologies, we prioritized concatenated ML trees in the main text, with ASTRAL results provided in the Supplementary Materials for transparency.

4) Finally, I think it is debatable whether the approach of combining phylogeny and morphology to create a classification is truly novel as claimed as similar approaches have been taken before (e.g., see Muñoz-Rodríguez et al. 2023, <https://doi.org/10.1002/tax.12887>) and suggest that any such claims should be toned down or removed.

Response 5: We agree and have removed the claims, as well as the associated Figure 6 and discussion in the main text.

I also attach the main and supplementary files with additional minor points for consideration by the authors.

I hope that these comments will be useful for further improving the quality of the paper and to present these significant achievements in the most appropriate way.

Response 6: We thank the reviewer for the detailed annotations in the main and supplementary files. All points have been systematically addressed below, with revisions implemented per the attached tracked-changes documents.

Line62: Compared to small ones, the taxonomy of big plant genera (with more than 500 species) are more challenging due to their greater number of species, extensive distributions, and complex variations¹⁸⁻²⁰.

Q1: "variations" is quite vague – please state more clearly what you mean by this.

Response 7: We have rephrased the passage as follows: "Their phylogeny and taxonomy are uniquely challenging: beyond the sheer number of species, they exhibit extensive (often intercontinental) distributions and frequent rapid radiations (e.g. *Astragalus*³), collectively complicating phylogenetic resolution and taxonomic completeness^{1,2,4}" (Lines 44–47).

Q2: This statement appears almost trivial – of course it is more challenging to work on a larger group than on a smaller group. In addition, the definition of "big" as > 500 species is arbitrary. I suggest revising this sentence.

Response 8: We have rephrased the passage as follows: "Big plant genera – often operationally defined by thresholds such as ≥ 500 species^{1,2} – account for ca. 25% of all flowering plant species²" (Lines 41–42). We hope that you will find these revisions satisfactory.

Line74: "Temperate genera, which are mostly emerging after the mid-Cenozoic and often associated with unique geological and environmental events such as the uplift of the Tibetan Plateau^{40,41}, global cooling⁴², grassland expansion^{43,44}, and desertification in Asian interior⁴⁰, exhibit higher diversification rates than the tropical ones^{45,46}. These genera harbor distinct lineages⁴⁷ and valuable resources⁴⁸, making them uniquely important. Consequently, completing the phylogeny and taxonomy of temperate big genera is an urgent priority for the botanical community."

Q3: This statement seems too absolute – there will certainly be some tropical genera with higher diversification rates than temperate genera. Suggest adding "often" or similar.

Response 9: Thank you for your suggestion! We have added the word "often" and rephrased the passage as follows: "Originating predominantly during mid-Cenozoic geo-climatic upheavals—such as global cooling⁹, grassland expansion¹⁰, and desertification¹¹—temperate genera often exhibit accelerated diversification rates¹² and acquisition of novel morphological or physiological adaptations to new niches (e.g., cold and/or aridity)^{12,13}" (Lines 55–59).

Q4: Can you please expand on this? It could be said of almost any genus that it harbors distinct lineages, and thus this statement appears trivial. Please provide some additional information on why or how big genera are uniquely distinct.

Response 10: We have rephrased the passage as follows: "These Cenozoic-forged evolutionary innovations position the temperate big genera as unparalleled systems for studying adaptive evolution under environmental upheavals. Their distinct metabolites—such as the antimalarial dominance of *Artemisia annua*'s artemisinin¹⁴, which has replaced tropical *Cinchona*-derived quinine in the past decades—reveal untapped potential for bioprospecting¹⁵" (Lines 59–63). We hope that you will find these revisions adequate and satisfactory.

Line85: "Although it has not yet been applied to big plant genera, the construction of a gigamatrix, which integrates Sanger sequencing data (numerous species and few sequences) with high-throughput DNA sequencing data (some species and many sequences)^{6,54-56}, offers a promising strategy for building a comprehensive and robust phylogeny."

Q5: I think this statement is somewhat problematic. "Gigamatrix" approaches combining different sources of data have been used before for even larger groups, e.g. the entire palm family with ca. 2500 species (Faurby et al. 2016). Instead of making a claim that something like this has never been done, I would suggest instead highlighting other cases where such an approach has been used successfully.

Reference:

Faurby, S., Eiserhardt, W.L., Baker, W.J., Svenning, J.C., 2016. An all-evidence species-level supertree for the palms (Arecaceae). *Mol Phylogenet Evol* 100, 57-69.

Response 11: We thank the reviewer for pointing us to this reference. We have rephrased the passage as follows: "Some pioneer attempts have been made to resolve the phylogenies of species-rich taxa by using Sanger sequencing data¹⁶. However, resolution in these phylogenies is restricted due to limited evolutionary signals inherent to such data. Recent advances in high-throughput DNA sequencing have facilitated the resolution of complex phylogenetic relationships by utilizing extensive evolutionary signals across entire genomes¹⁷. How to balance taxonomic coverage and genomic depth for a comprehensive and robust phylogeny is a big challenge. The construction of a gigamatrix^{18,19}, which integrates high-throughput DNA sequencing data (some species and many sequences) with Sanger sequencing data (numerous species and few sequences), offers a promising strategy to address this gap²⁰" (Lines 65–73). We hope that you are satisfied with this revision.

Line102: "A comprehensive and global phylogenetic framework with high resolution and a natural infrageneric taxonomy are crucial for understanding and using this economically important genus, but a robust phylogeny and a complete taxonomic

hypothesis for each species of *Artemisia* remain missing even after many efforts^{49,69-81} .

Q6: Please explain what you mean by "natural" – this leaves very wide room for different interpretations.

Response 12: We acknowledge "natural" may be ambiguous in this context. Following Muñoz-Rodríguez et al. (2023 *Taxon* 72, 1201–1215), we intended it to mean a taxonomy fulfilling four criteria: monophyly, resolution, diagnosability, and completeness. The text now reads: "A high-resolution global phylogenetic framework and comprehensive infrageneric taxonomy are crucial for understanding and utilizing this scientifically and economically important genus" (Lines 85–87), removing potential ambiguity while maintaining taxonomic rigor.

Line107: The infrageneric taxonomy of *Artemisia* predominantly relies on the morphology of the capitula and leaves as well as life forms^{74,82,83} .

Q7: Add "existing"?

Response 13: We agree and have added "existing", revising the sentence to: "The existing infrageneric taxonomy of *Artemisia* predominantly relies on the morphology of the capitula and leaves as well as life forms" (Lines 89–90).

Line129: "Here we first reconstructed phylogenetic frameworks for *Artemisia* utilizing low copy nuclear genes and complete plastomes respectively, followed by a comparative analysis to provide comprehensive phylogenetic evidence from different genomes. Then, by integrating the low copy nuclear genes and two ribosomal DNA markers, we reconstructed a phylogeny for *Artemisia* with approximately 70% of this species. Furthermore, we inferred the evolutionary trajectories of 19 morphological characters traditionally or potentially significant for taxonomy of this genus. Based on these findings, we propose a revised global taxonomy for *Artemisia*, including 8 subgenera, 24 sections, and 501 out of the 504 accepted species with taxonomic placements. This study provides a comprehensive framework for advancing our understanding of the evolution, ecology, and sustainable use of *Artemisia*, and introduces a novel approach for taxonomic research on big genera in the genomic era."

Q8: Please provide the number of nuclear genes here.

Response 14: We have specified the exact number of nuclear genes and revised the text to "integrating 202 low-copy nuclear genes and two rDNA markers" (Line 112) for enhanced precision.

Q9: Please be precise here - 346 out of 504 species would be 68.7%.

Response 15: We have refined and updated the wording to "78%" based on our new results (394 out of 505 species, Line 111).

Q10: I think it is debatable whether this approach of combining phylogeny and morphology to create a classification is truly novel - I think this claim should be removed. For example, see Muñoz-Rodríguez et al. 2023.

Muñoz-Rodríguez, P., Wood, J.R.I., Wells, T., Carruthers, T., Sumadijaya, A., Scotland, R.W., 2023. The challenges of classifying big genera such as *Ipomoea*. *Taxon* 72, 1201-1215.

Response 16: Thanks for pointing this out. We have revised the wording to "introduced a new case on how to tackle taxonomic challenges in big genera in the genomic era." (Lines 121–122).

Line 141: "

Results

The most comprehensive molecular phylogenetic sampling

We successfully obtained genome-skimming data for 228 species of *Artemisia* and its allies, and then constructed three datasets: 1) 216 nuclear low-copy genes (hereafter NLC data); 2) the comprehensive plastome (hereafter plastome data); and 3) the ITS and ETS regions of ribosomal nuclear DNA (hereafter NR data). We also got ITS and ETS sequences of 138 species without genome-skimming data (Table S2)."

Q11: I think there is a risk of over-selling your achievements as the paper is almost exclusively based on existing published sequence data. I suggest to tone this down.

Response17: We merged the original two sections related to *Artemisia* phylogeny into "An expanded phylogenetic framework incorporating 78% *Artemisia* species" (Line 124) to reduce redundancy and to tone the statement down.

Our key advances over previous studies are summarized as follows:

Generated genome-skimming data for additional 96 *Artemisia* species, increasing the species coverage of *Artemisia* from 41% (205 species; Jiao et al., 2023 *Ann. Bot.* 131, 867–883) to 78% (394 species; this study) (Line 145; Fig. 2).

Acquired 202 nuclear low-copy genes from 314 species (298 *Artemisia* + 16 allies), surpassing Jiao et al. (2023)'s SNP data for 215 species (205 *Artemisia* + 10 allies) (Line 131; Fig. S2).

Obtained ITS/ETS sequences for 414 species (394 *Artemisia* + 20 allied species, 80% newly generated), exceeding Malik et al. (2017 *Taxon* 66, 934–952)'s 266 species (263 *Artemisia* + 3 allied species) (Lines 132 and 464–469; Fig. S4).

Reconstructed the first plastome phylogeny for 314 species (298 *Artemisia* + 16 allied species, 59% representation) (Lines 196–204; Figs. 3, S6).

We think these advancements justify the term "comprehensive" where appropriately qualified. We hope that you will find these explanations and revisions satisfactory.

Q12: Suggest deleting – redundant with "obtained"

Response18: We deleted "obtained" and rephrased this sentence as follows: "We assembled complete plastomes for 314 species (298 *Artemisia* + 16 allied species, plastome dataset) from our genome-skimming data..." (Lines 194–195).

Q13: This reads a bit clunkily – can you rephrase? One other option could be e.g. "entire plastomes"

Response 19: Thank you for your helpful suggestions! We have rephrased the text to: "We assembled complete plastomes for 314 species (298 *Artemisia* + 16 allied species, plastome dataset) from our genome-skimming data..." (Lines 194–195) to improve clarity and flow.

Line170: "**Fig. 2. Phylogenetic discordance and reticulate evolution in *Artemisia*.** a) Tanglegram illustrating discordance between the Maximum likelihood phylogenetic trees obtained from 216 low-copy nuclear genes (NR data, left) and plastome data (right)."

Q14: Above, you abbreviate low-copy nuclear genes as NLC, whereas you use NR for two nuclear ribosomal regions. This makes it impossible to know what you are referring to. This issue also occurs in other parts of the text, please check and correct throughout.

Response 20: We standardized abbreviations throughout: NLC for low-copy nuclear genes; NR for nuclear ribosomal regions. This resolves prior inconsistencies. We appreciate your meticulous review.

Line184: "The primary cytonuclear discordance recovered in *Artemisia* is summarized below."

Q15: There is considerable emphasis on discordance but relatively little text on the commonalities or the topology of the trees. This has the unfortunate effect that the reader learns about any problematic issues in detail, but only little about the actual relationships that the authors have recovered. To address this issue, I think it would make more sense to present the results of the "main analysis" first in some detail, and only afterwards describe how other trees differ from this. Given that you describe the topology of the gigamatrix tree below in section 3., the easiest solution to achieve this might be to swap sections 2 and 3.

Response 21: Thanks for your wonderful suggestions. We have swapped sections 2 (Lines 193–263) and 3 (Lines 124–192), and the figure sequence has been updated accordingly. We hope this will increase its clarity.

Line213: "**The most comprehensive phylogeny of *Artemisia* to date**

Our phylogeny based on gigamatrix data (hereafter referred to as GM data) comprised 346 *Artemisia* species (ca. 70% of the total number, Fig. 3, S8), along with 20 outgroup species. Among these, ca. 60% (228/366) had both NLC and NR data available, while ca. 40% (138/366) were represented solely by NR data."

Q16: Is this based on the concatenated dataset or the ASTRAL tree?

Response 22: To eliminate ambiguity, we have explicitly specified the reconstruction method in the revised manuscript: "the concatenated gigamatrix (GM dataset) comprising the NLC dataset and the NR dataset" (Lines 129–130); and "Using maximum likelihood (ML) analysis of the GM dataset, we reconstructed a fully resolved phylogeny of *Artemisia*" (Lines 139–140).

Q17: Please give precise percentages here, e.g. in this case 62% and 38% or 62.3% and 37.7%. No need to do such coarse rounding, and in this case, it even under-sells the quality of your data!

Response 23: We have replaced all approximate numbers in the manuscript with precise values (e.g., Lines 133, 134, and 145–146).

Line244: "**Evolutionary trajectories of macro- and micro-morphological characters and the sectional taxonomy of *Artemisia***

Twelve readily observable macromorphological characters, such as life form, plant height, synflorescence type, capitula shape, and leaf type, were meticulously documented for all sampled *Artemisia* species through field observations, specimens examinations, and literature review (Table S3)."

Q18: Suggest deleting – it is a general expectation that scientists work meticulously.

Response 24: We deleted "meticulously" and rephrased the text to: "Thirteen macromorphological characters, including life form, plant height, synflorescence, capitulum, and leaf morphology (Tables S3 and S4) were analyzed based on the ML tree derived from GM dataset (Figs. 4 and S12)." (Lines 268–270).

Line256: "Notably, aside from the characters previously employed in subgeneric taxonomy (e.g. capitula type)⁶⁹, leaf type was revealed to be an important one for the first time, as it had the strongest phylogenetic signal (Pagel's $\lambda = 0.96$, Table S3) and thus could be useful in delineating subgenera and sections within *Artemisia*."

Q19: This is quite a strong claim, and also somewhat surprising. Please at least add references to previous classifications.

Response 25: We appreciate your alerting us to this issue. The text has been revised to: "Notably, aside from capitulum type, which was previously employed in

infrageneric taxonomy²¹⁻²³ ($\lambda = 0.93$, $K = 1.99$; Figs. 4 and S12, Supplementary Data 1 and 4), leaf type, a novel character introduced in this study, was identified as a key infrageneric taxonomic character due to its strong phylogenetic signal ($\lambda = 0.99$, $K = 1.50$; Fig. S12, Table S3, Supplementary Data 4). " (Lines 276–280).

Line262: "**Fig. 3. Concatenated ML tree based on the gigamatrix of 366 species of *Artemisia* and its allies.**

Branches with a bootstrap support value ≥ 75 are highlighted in bold. The tree with branch length and detailed support values is shown in Fig. S8. The outer and inner colored curved lines represent the subgenus and section, respectively. The outermost images illustrate the habitat and the morphological (capitula and florets) diversity within *Artemisia*. "

Q20: Why did you choose to present the concatenated tree instead of the ASTRAL tree? Generally, I think ASTRAL would be preferable because it accounts for incomplete lineage sorting and gene tree conflict.

Response 26: In the revised manuscript, we added a topological comparison of nuclear trees (ASTRAL and concatenated ML) in Fig. S3, with an explanation for presenting concatenated ML trees in the main text (Lines 141–147), summarized as follows:

Data heterogeneity constraints: The gigamatrix combines nuclear genomic data (abundant sequences, fewer species) and ribosomal DNA (more species, sparser sequences), creating heterogeneity that rendered multi-species coalescent analysis (ASTRAL) computationally infeasible and statistically unreliable for the full dataset (Portik et al., 2023 *Mol. Biol. Evol.* 40, msad109; Lines 496–498).

Topological robustness: While both concatenated and ASTRAL analyses were performed on NLC dataset (Figs. S2 and S5), concatenated trees exhibited more stable and robust support for major *Artemisia* clades compared to coalescent-based methods (mean 99% vs. 58% in ASTRAL, see Fig. S3 for comparison). Besides, the backbone of the ASTRAL tree is consistent with concatenated ML tree, differing in that the genus *Artemisiella* was embedded within *Artemisia*, and both *A. subg. Dracunculus* and *A. subg. Ponticae* were polyphyletic (Fig. S3), with discordant nodes having low local posterior probability (LPP < 0.5 , Fig. S3), indicating minimal conflict. Given this congruence and the superior robustness of concatenated topologies, we prioritized concatenated ML trees in the main text, with ASTRAL results provided in the Supplementary Materials for transparency.

Thank you for prompting this clarification.

Q21: I think the images on the outermost circle don't work very well because the images don't align well with the clades they represent. In addition, different characters appear to be shown in the images mixed together, making it difficult to

identify the important differences. I would suggest reducing the number of images (e.g., I think the habit pictures could be removed) and making sure that they are aligned well with the clades.

Response 27: Thank you for your constructive comment on Figure 3 (renumbered as Fig. 2 in revised manuscript). Floret morphology photos highlighting capitulum types—a key characters in *Artemisia*—now replace habitat photos for representative species across all 24 sections. In the revised Figure 2, dashed lines connect these to corresponding species, ensuring clear visual association (Lines 837–840). We hope that you will find these revisions satisfactory.

Line294: The results revealed that Pagel's λ^{93} and Blomberg's K^{94} values for corolla shape and style morphology of the floret were greater than or close to 1 (Fig. S12, Table S5).

Q22: Please explain what this means – readers not familiar with these statistical measures will struggle to make any sense of it.

Response 28: We have added an explanation of phylogenetic signals in Methods section: "Phylogenetic signal denotes..." (Lines 566–569). We hope this will increase its clarity for the readers.

Line363: Species of *A.* subg. *Pectinatae* are throughout in the plastome tree of *Artemisia*, with five sampled species nested within five distinct subgenera (Fig. 2a, S3).

Q23: Check grammar – something missing here.

Response 29: Revised to: "Species of *A.* subg. *Pectinatae* were scattered across the plastome tree, with five sampled species nested within five distinct subgenera (Fig. 3a)." (Lines 244–246).

Line 372: Given the prevalence of hybridization events, reliance on maternally inherited plastome data alone is insufficient to elucidate the evolutionary history of *Artemisia*, especially with the occurrence of chloroplast capture events, which can mislead the reconstruction of organismal phylogeny⁹⁶. Consequently, our current taxonomy of *Artemisia* primarily relies on phylogenies derived from nuclear data. Nonetheless, we acknowledge that plastome is an essential component of plant genome. Through comparative phylogenetic studies, plastome data can help to provide new insights into the evolutionary history of morphological and ecological characters^{51,97,98}.

Q24: These are important general points but difficult to find as they are buried within the three case examples. Please restructure to make these points more prominent.

Response 30: We restructured this section as the concluding paragraph (Lines 252–262) to emphasize: Hybridization and chloroplast capture occur in particular

Artemisia lineages, causing cytonuclear discordance that limits plastome-based phylogenies. Chloroplast capture can mislead phylogenetic inference, complicating relationship interpretation. Thus, *Artemisia* taxonomy primarily relies on nuclear phylogenies for robust resolution. Plastomes remain essential genomic components, and their comparative analysis provides valuable complementary evolutionary insights when integrated with nuclear data. We hope that you will find these revisions adequate and satisfactory.

Line380: **2. The most comprehensive phylogeny provides new insights into the systematic and evolution of *Artemisia***

Q25: I think this should be the first section in the discussion – it makes more sense to start with general points and only afterwards delve into details such as discordance.

Response 31: Very good suggestion! We relocated the phylogeny discussion to the opening section per the reviewer's suggestion (cf. Response 21) to enhance logical flow.

Line390: 1) *Ajaniopsis*, an endangered genus endemic to alpine scree slopes of Tibetan Plateau was unexpectedly found to be deeply nested within *Artemisia*⁹⁹ (Fig. 3, S8). Its unique combination of characters (apically pilose florets, 5- or 6-ribbed achenes, and corymbose synflorescences)¹⁰⁰ may be the result of convergent evolution, as these characters also occurred in other species of *Artemisia* (e.g. *A. albicans*, *A. inaequifolia*, *A. glacialis*, and *A. glomerata*) which are geographically distant.

Q26: Are they also phylogenetically distant?

Response 32: Yes, they are phylogenetically distant. Revised for precision (Lines 179–181): "as these characters also occur in other species of *Artemisia*—such as *A. albicans* (subg. *Tridentatae*, western North America) and *A. glacialis* (subg. *Absinthium*, European Alps)—that are both phylogenetically and geographically distant (Fig. 2)".

Line410: **3. Complete sectional taxonomy of *Artemisia***

Despite numerous challenges, pursuing a new taxonomic framework that identifies all recognized infrageneric taxa through morphological characters and robustly validates them via molecular phylogeny is worthwhile. This framework can be further enriched by incorporating additional evidence such as geographic distribution and ecology.

Q27: I agree, but I think this sentence needs to be revised. Just stating that something "is worthwhile" is a subjective statement. To make it more comprehensible to others, please add why you think is worthwhile.

Response 33: We agree and have revised to emphasize objective scientific value: "Our study provides a robust baseline for future studies of this ecologically and

economically important big genus, such as local revisions, bioprospecting, conservation, and comparative evolutionary analysis." This now appears in the Conclusion (Lines 361–363).

Q28: I wonder whether it would be better to discuss this in the conclusion as you have not acted on this?

Response 34: This point has been addressed by relocating this sentence to the Conclusion (Lines 361–362), as tracked in the revised manuscript.

Line435: In this revised taxonomical framework, 501 accepted species are classified into eight subgenera and 24 sections, ensuring that each species has a testable taxonomic and evolutionary hypothesis, with only three species being left as unplaced (Table S6, SI).

Q29: Please add some details on these three species and why it was not possible to get any data for them. It is somewhat frustrating that you managed to place 99.4% of all species but that just these three remain unplaced. Highlighting them may help them receiving additional attention in the future.

Response 35: Implemented per suggestion: Added main text explanation for unplaced species (Lines 330–332) while preserving supplementary details.

Line441: This approach decomposed a large and complex genus (504 species) into many small sections, each with an average of less than 30 species (mean 19, minimum 1, maximum 119).

Q30: Given that the mean is 19, you should change this "an average of less than 20 species", or just directly state that the average is 19 species.

Response 36: Revised to: "This new taxonomic framework decomposed this big and complex genus (505 species) into 24 smaller sections (mean 19 species/section; range 1–119)" (Line 336–337).

Line467: Based on this framework, we propose a new workflow for revising the taxonomy of big genera in the genomic era (Fig. 6).

Q31: As mentioned above, I think it is debatable whether this approach of combining phylogeny and morphology to create a classification is truly novel - I think this claim should be removed or at least toned down. For example, see Muñoz-Rodríguez et al. 2023.

Muñoz-Rodríguez, P., Wood, J.R.I., Wells, T., Carruthers, T., Sumadijaya, A., Scotland, R.W., 2023. The challenges of classifying big genera such as *Ipomoea*. *Taxon* 72, 1201-1215.

Response 37: Thanks for pointing this out. We have revised the wording to "This work serves as an exemplary case for taxonomic research on big genera in the genomic era" (Lines 367–368). We hope you are satisfied with this revision.

Line489: **Conclusions**

Q32: The conclusion in its current form is essentially a re-write of the abstract, with little additional information given.

I would suggest to fully revise the conclusion. For example, you could focus more on what can be learnt from this study for similar projects in other clades, and what the next steps for *Artemisia* should be.

Alternatively, you might consider deleting the conclusion, as section 4 above lays this out quite nicely.

Response 38: We have restructured the Conclusions (Lines 353–368) to incorporate the suggested improvements. We hope that you are satisfied with this revision.

Line532: We obtained ~3 Gb of data for each sample with paired-end libraries. The average length of the generated reads from silica gel-dried and herbarium specimens was 150 and 100 bp, respectively.

Q33: Can you double-check this? Silica material usually yields longer reads than herbarium material, so I am surprised to read that silica yielded shorter reads. Revising this sentence may help to avoid ambiguity.

Response 39: We clarified the text: "The average read length was 150 bp for silica-gel-dried leaves, and 100 bp for herbarium specimens" (Lines 405–406).

Line541: Sequencing was carried out on the Illumina HiSeq 2500 platform, employing 100-150 bp paired-end sequencing with a through put of 6 Gb. The RNA sequencing raw reads were cleaned using SeqyClean¹⁰⁴ to trim poly-A/T tails and terminal nucleotides on a sliding window of 10 bp that averaged a Phred score of 10 or less.

Q34: Please provide version numbers throughout – this also applies to other methods mentioned, e.g. GoldFinder.

Response 40: We have added version numbers for all the software.

Line593: All samples that were potentially misidentified (i.e. those occupying highly unlikely phylogenetic positions, such as a different subgenus than expected) were removed from further analyses⁴¹.

Q35: Please state how many samples were excluded in this step.

Response 41: Actually only one sample—initially labeled *Artemisia palmeri*—was excluded due to incongruent phylogenetic placement. See Response 42 for details.

Q36: While I understand your motivation, I think this needs to be explained in a bit more detail as there is a risk of confirmation bias. Please state what your subgenus expectations were based on, and whether also other factors, e.g. morphology, were considered? Can you exclude the possibility that the unexpected phylogenetic placements were in fact correct?

Response 42: Thank you for your thoughtful query. Only one sample (originally labeled as *Artemisia palmeri*, re-identified as *A. absinthium*; voucher: T. G. Gao 7613, PE) was excluded due to conflicting phylogenetic placement. Our initial genomic data placed it in *A. subg. Absinthium*, contradicting morphological evidence (Shultz, 2006 *Flora of North America* 503–534) and prior phylogenies (Garcia et al., 2011 *Am. J. Bot.* 98, 638–653; Hobbs & Baldwin, 2013 *J. Biogeogr.* 40, 442–454), which uniformly support its position in *A. subg. Artemisia*. This discrepancy suggested potential mislabeling, prompting its intended exclusion. However, an oversight retained it in the original ML tree (Fig. 3 in our initial submission). To resolve this, we incorporated GenBank sequences of *A. palmeri* (Accession numbers: HQ019052, JX051740), which confirmed its placement in *A. subg. Artemisia* (Fig. 2 in the revised manuscript). Quality control of other samples from the same batch identified no similar issues. The mislabeled sample was removed in the revised manuscript to adhere to our stated strategy of one sample per species. As this issue is now resolved, we omitted the related discussion to streamline the manuscript. We hope that you will find these revisions satisfactory.

Line610: Before running ASTRAL III, we removed poorly resolved topologies in gene trees. Branches with bootstrap support (BS) $\leq 20\%$ were collapsed using the ‘nw_ed’ function in Newick Utilities¹¹⁷.

Q37: I assume you collapsed branches in the gene trees? Please add this.

Response 43: Implemented per suggestion: Added methodological clarification (Lines 511–513): "Before running ASTRAL III, we collapsed branches with bootstrap support (BS) $\leq 20\%$ in gene trees using the ‘nw_ed’ function in Newick Utilities⁷⁷".

Line643: We collected data for 12 macromorphological characters (Table 1). In addition to the characters previously used for subgeneric taxonomy⁶⁹, such as life form, capitula type, inflorescence type, leaf shape, and leaf size, we identified others, including plant height, capitula diameter, leaf length-width ratio, leaf segment length and leaf segment width.

Q38: Please replace this with "leaf area:" throughout to avoid confusion.

Response 44: We appreciate your critical question regarding leaf measurements. Our initial approach to leaf size classification followed *Manual of Leaf Architecture* (Ellis

et al., 2009), which defines leaf size as leaf area, either directly measured or approximated by length \times width \times 0.75 (Cain & Castro, 1959 *Manual of vegetation analysis*).

As you correctly noted, the high variation in leaf dissection within *Artemisia* (empirical correction factor range: 0.03–0.85) invalidates a universal 0.75 factor. To address this, we:

Retained "leaf size" as a descriptive term for overall dimensions, independent of dissection (Lines 554–556);

Added precise "leaf area" measurements using ImageJ (Abràmoff et al., 2004 *Biophotonics Int.* 11, 36–41), defined as one-side projected area per angiosperm standards (Perez-Harguindeguy et al., 2013 *Aust. J. Bot.* 61, 167–234; Lines 556–557).

This revision aligns with methodological rigor in leaf morphology studies in angiosperm, ensuring accuracy while acknowledging within-genus variation. We hope that you will find these revisions adequate and satisfactory.

Q39: Table 1 shows that all characters are coded as qualitative characters, even quantitative characters such as length or width. The division of a continuous range into discrete characters (e.g., big vs small) may influence the results, depending on which thresholds are selected. Please explain why you decided to code all characters as qualitative, and on which basis you decided into how many different categories a quantitative range should be divided, and where the thresholds between the categories should be.

Response 45: We coded all characters as qualitative to enable standardized taxonomic descriptions and facilitate diagnostic key construction, aligning with established practices in plant taxonomy (Stuessy, 2009 *Plant Taxonomy*). Additionally, discrete coding helps reduce the impact of measurement error on analysis (Thiele, 1993 *Cladistics* 9, 377–385).

Initially, the number of categories and threshold values were determined based on our taxonomic expertise and field experience developed in the past decade. In revision, we incorporated a data-driven approach: Gaussian Mixture Model (GMM) analysis was applied to determine optimal category numbers via the Bayesian Information Criterion (BIC), with thresholds set at cluster boundaries (Tiburtini et al., 2024 *bioRxiv*, Lines 560–563). The scripts used here are available at figshare data repository (<https://doi.org/10.6084/m9.figshare.28164335>). Our updated analysis, detailed in Table S4, reinforces the conclusions presented in our original submission. Thank you very much for your good suggestion!

Line652: Based on the obtained *Artemisia* gigamatrix tree (Fig. 3), we used the "APE" package⁹² of RASP v4.2⁹¹ to reconstruct the ancestral states of the 12

characters based on the maximum likelihood method. Among them, leaf size follows the standard of Webb¹²²: An approximation of leaf size can be obtained by measuring the length and width of the leaf and multiplying the length \times width \times 3/4¹²³.

Q40: I think you should use "area" here instead of "size", because "size" is somewhat ambiguous.

Response 46: We appreciate this suggestion. "Leaf area" has been added as a distinct character in our analyses. The measurement focus and application of both characters are now clarified in the Methods section (Lines 554–557) to eliminate ambiguity.

Q41: This leaf size calculation appears flawed given that the leaves of *Artemisia* are often highly dissected, as shown in Fig. 4, and often appear to occupy only half or less than half of length \times width. I think it would be more convincing to base this multiplying factor on actual measurements in *Artemisia*. Alternatively, have you conducted leaf area measurements to confirm that the factor used is a reasonable approximation of the actual areas?

Response 47: We appreciate this critical observation. Given the extensive variation in leaf dissection across *Artemisia* (as illustrated in Fig. 4), we did not provide a universal correction factor for the entire genus, and instead introduced actual measured values of leaf area as a new morphological character (named as leaf area) in the revised manuscript (Lines 556–557). These measurements, which account for dissected leaf geometry, are documented in Supplementary Data 4. For further details, please see Response 44 above.

Line673: Based on this pruned GM tree, ancestral states and phylogenetic signals of these seven microcharacters were estimated using the method described above. Since this was the first global analysis of these microcharacters, data for these unavailable species could not be obtained from previous studies.

Q42: I am not sure whether I fully follow this – even if it is the first such global analysis, it is possible that taxa not included in your study were previously included in other, more localized morphological studies or taxonomic treatments. Are you in fact saying that these characters have never been studied / measured before?

Response 48: We appreciate the opportunity to clarify. Our previous description of micromorphological characters may have been ambiguous. While some taxonomists (e.g., Poljakov, 1961 *Trans. Inst. Bot. (Alma - Ata)* 11, 134–178) recognized their taxonomic potential, comprehensive global studies remain extremely limited—largely confined to isolated species and few characters (Jiao et al., 2019 *Syst. Bot.* 44, 424–432; Park et al., 2010 *Korean J. Plant Taxon.* 40, 27–42 as far as we know). This scarcity stems from technical challenges: *Artemisia capitulum* is typically < 5 mm in diameter, containing > 20 florets with microscopic internal structures (e.g., anthers). Our study presents the first systematic global analysis of these characters. The revised text now states: "Although some taxonomists^{37,38} realized the taxonomic potential of

these characters, to our knowledge, none have been systematically studied^{38,58}." (Lines 295–297).

Line692: The species with insufficient evidence were treated as unplaced and listed separately. For the conflicting taxonomic treatments, we followed our own based on our observations in the field and herbaria, as well as the latest phylogeny provided here. We acknowledge that since the aim of the current study is not to delimitate species boundaries, our treatment herein is provisional. Much more work in the field, herbaria, and labs is needed in the future.

Q43: Do you mean insufficient morphological or genetic evidence, or both? Please add this.

Response 49: We mean that both morphological and molecular evidence are insufficient. This sentence has been rephrased as follows: "Three species (*Artemisia avarica*, *A. dipsacea*, *A. galinae*) remain unplaced due to insufficient morphological and molecular evidence, primarily stemming from ambiguous protologues and specimen inaccessibility" (Lines 330–332).

Q44: Is a noun missing here? Not clear what you followed.

Response 50: We appreciate the reviewer's observation. The sentence has been revised to clarify: "we proposed a complete sectional taxonomy of *Artemisia*, identified diagnostic morphological characters for each subgenus and section, and developed an identification key to all of them (Supplementary Data 3) based on the phylogenies from genomic data (Fig. 2) and analysis of morphological characters (Figs. 4 and 5)." (Lines 321–324).

Q45: I think this would not even be possible, given that only one sample per species was analysed.

Response 51: We acknowledge the single-sample limitation precludes species delimitation analysis. Consequently, we revised the text to state: "Notably, our single-representative-per-species approach does not address species delimitations in *Artemisia*, and thus this species treatment herein remains provisional, consistent with the inherently dynamic nature of taxonomic studies." (Lines 348–351). This sentence was moved to the Results and discussion section.

Q46: Treatment of what? Treatment of species?

Response 52: "Treatment" refers to our taxonomic treatment of *Artemisia* species. For precision, we rephrased this sentence to: "this species treatment herein remains provisional" (Line 350).

Q47: I wonder whether it would make more sense to move these two final sentences in revised form to the conclusion as they are not really "methods"?

Response 53: We agree and have relocated them to the Results and discussion section (Lines 349–351).

Supplementary information

Line61: "**Unplaced species:** *Artemisia avarica* Minatul., *A. dipsacea* Krasch., *A. galinae* Ikonn.

Notes: For these three species, we have not obtained sufficient morphological or molecular evidence to classify them into any subgenus or section due to either the obscurity of the protologues or the inaccessibility of any specimens. Hence, further research on their taxonomy is required. "

I think it would be good to place this information in the main manuscript to highlight these three species as requiring taxonomic attention.

Response 54: We have incorporated the information about the three unplaced species into the Results and discussion section (Lines 330–332).

Line89: "

- 7. Leaves small; blades filiform, linear..... *Artemisia* subg. *Pectinatae* sect. *Pectinatae*
- 7. Leaves medium; blades oblong. *Artemisia* subg. *Pectinatae* sect. *Hedinianae*"

Leaf sizes such as "small" etc. need to be defined here, otherwise it is highly subjective!

Response 55: We addressed this by establishing quantitative thresholds to define "small", "medium", and "large" leaf categories prior to the identification key. Additionally, we systematically revised the entire key to replace all ambiguous terms with clear, operational definitions (see details in the Supplementary Data 3 of the revised manuscript).

Dr Benedikt Kuhnhäuser
Royal Botanic Gardens, Kew

Reviewer #4 (Remarks to the Author):

This study represents an updated addition to the understanding of the evolutionary relationships and classification of the genus *Artemisia*. It takes on a previous study by Jiao and collaborators from 2023, in which they created a large genome skimming dataset to study the evolutionary relationships in this mega diverse genus. The dataset is recycled in this new manuscript, but data exploitation is based on an alternative

approach to use genome skimming data, extracting low copy genes, chloroplast data and nrDNA loci.

Response 1: We appreciate your insightful comments. Through the collaborative efforts of our international team, we have incorporated newly generated genome-skimming data for 96 additional *Artemisia* species, increasing species coverage from the previously reported 41% (Jiao et al., 2023 *Ann. Bot.* 131, 867–883) to 78% in the revised manuscript. The expanded dataset not only confirms our original conclusions but also strengthens the comprehensiveness and robustness of the global phylogenetic and taxonomic framework for *Artemisia*.

Overall, the manuscript is well written and cohesive, the results are strong and thoroughly discussed, but I have some comments for the authors to consider.

-In my opinion, this is an interesting approach, but I somehow miss throughout the text a justification about why following this new approach is essential for understanding *Artemisia*, if the results from this study mirror to a large extent those from Jiao et al 2023, with most of the taxonomical implications on the new classification following those from the study from 2023. I believe adding an statement emphasizing on this point will strengthen the relevance and impact of this new study, especially for a multidisciplinary journal such as Nat. Comms.

Response 2: We appreciate the opportunity to clarify how our study advances *Artemisia* phylogeny and taxonomy. Relative to Jiao et al. (2023 *Ann. Bot.* 131, 867–883), our study:

1) Increased the species coverage in the phylogenetic analysis of *Artemisia* from 41% (205 species; Jiao et al., 2023) to 78% (394 species; this study) (Lines 143–145; Fig. 2) in the revised manuscript.

2) Newly obtained 202 nuclear low-copy genes of 314 species (298 *Artemisia* + 16 allied species) [vs SNP data of 215 species (205 *Artemisia* + 10 allied species)] to reconstruct the phylogeny of *Artemisia* (Lines 131 and 380, Figs. 3 and S2) in the revised manuscript.

3) Generated plastome data for 314 species (298 *Artemisia* + 16 allied species) (vs only SNP data from nuclear genome) for the first time, and reconstructed a comprehensive plastome phylogeny of *Artemisia* with high species representativeness (59%), thus provided multi-genomics evidence for the evolution of *Artemisia* (Lines 194–196; Figs. 3 and S6).

4) Studied 20 (vs 6) morphological characters that can potentially be used for *Artemisia* infrageneric taxonomy (Figs. 4, 5, S12 to S14), including 7 micromorphological characters subjected to the first global analysis (Lines 542–546, 571–572).

6) Employed the gigamatrix approach that integrates high-throughput DNA sequencing data (some species and many sequences) with Sanger sequencing data (numerous species and few sequences), offered a new strategy to tackle the long-standing challenges in reconstructing a robust global phylogeny for a big genus (Lines 132–133, 139-140).

To facilitate clear comparison, we compiled a table as follows:

Table 1. Comparisons between this study and Jiao et al. (2023).

		Jiao et al., 2023	this study
	Artemisia species coverage	205	394
Taxon Sampling	species with genome-skimming data	228	314
	number of species with complete plastome	0	314
Morphology	number of macromorphological characters studied	6	13
	number of micromorphological characters studied	0	7
	number of subgenera	8	8
	number of sections	0	24
Taxonomy	species with subgeneric placements	205	502
	species with sectional placements	0	502
Methodological improvements	Nuclear genome data	Single Nucleotide Polymorphism (SNP)	202 low-copy genes
	Cytoneuclear discordance analysis	No	Yes

Following your suggestion, we added some statements in the Results and discussion and Conclusions in the revised manuscript:

"This update included the establishment of 10 new sections and revised circumscriptions for 11 existing ones (Supplementary Data 3)" (Line 324).

"In total, 502 of 505 accepted *Artemisia* species were classified into 8 subgenera and 24 sections" (Line 327).

"The differences between the present taxonomy and previous taxonomies (most regional) were summarized in Supplementary Data 6" (Line 334).

"Our integrative analysis of genomic and morphological data resolves long-standing phylogenetic and taxonomic complexities in the big genus *Artemisia*, establishing a phylogenetically robust and comprehensive taxonomic framework that proposed testable hypotheses for nearly all recognized species. Our study, along with other large-scale investigations of tropical big genera (e.g. *Ipomoea*⁴), demonstrates that actively adopting new technologies, exploring novel morphological characters, and fostering global collaboration, can tackle these long-standing challenges, and even accelerate the process that might otherwise take decades or even centuries. Our study provides a robust baseline for future studies of this ecologically and economically important big genus, such as local revisions, bioprospecting, conservation, and comparative evolutionary analysis." (Lines 354–363).

By resolving long-standing biodiversity research challenges and establishing a robust phylogenetic-taxonomic framework for future *Artemisia* studies, we think our work will enable taxonomic revisions, bioprospecting, conservation, and evolutionary analyses of this ecologically and economically significant genus. Notably, our 2023 *Artemisia* taxonomy paper (Jiao et al., 2023 *Ann. Bot.* 131, 867 – 883) has been cited 31 times as of June 21, 2025, by researchers in phytochemistry, immunology, pharmacology, ecology, ethnobotany, biodiversity conservation, plant systematics, taxonomy, and molecular evolution, based on Web of Science. This interdisciplinary impact demonstrates our work aligns with *Nature Communications*' scope.

-I am also wondering why the SNPs information from Jiao et al. (2023) has not been used in this megamatrix approach, if at all possible, which might not be the case, but coding this information into the matrix could also provide additional strength to the analysis. The study provides, though, a significant increase of additional morphological characters from that previous study, which I have no doubts will be very useful to inform future taxonomical studies.

Response 3: Thank you for this insightful suggestion! We considered incorporating SNP data from Jiao *et al.* (2023 *Ann. Bot.* 131, 867–883) but opted against it for the following reason: the SNP calls were derived from whole-genome coding sequence (CDS) alignments that partially overlap with our nuclear gene dataset. Including both would have resulted in repeated use of shared loci, introducing potential bias in the subsequent phylogenetic inference. Instead, our focus on low-copy nuclear genes—augmented by novel morphological characters—provides a robust, non-redundant dataset sufficient to resolve deep species relationships with high confidence (Fig. 3d). We agree that the expanded morphological dataset strengthens the study's taxonomic utility, as highlighted in the revised Methods (Lines 542–546) and Results and discussion sections (Lines 316–318).

-I understand that sampling from highly diverse genera such as *Artemisia* is challenging and can be proved difficult. Certainly, the expanded sampling of 138 species in this new study refers mostly (if not all) to Genbank data on ITS/ETS nrDNA sequences, and to some extent means that all this additional phylogenetic

information relies on the analysis of two loci. Could this imply any sort of analytical bias when combined with the NLC dataset? One has to bear in mind that for multiple species the analysis relies on c. 900 bp whilst in other sequence information spans over 216 loci (c. 260000 bp). I would suggest the authors to comment on this. The phylogenetic approaches used in this manuscript are beyond my field of expertise, so I am not at all criticizing such take on, but consider a discussion on this point important to the readers to better understand any technical flaws arising from such strategies.

Response 4: We appreciate this critical insight regarding potential analytical biases when combining ribosomal DNA (ITS/ETS; ~900 bp) with low-copy nuclear gene dataset (NLC; 202 loci, ~260,000 bp). We address these concerns as follows:

- 1) High computational resource demands. Integrating large-scale datasets (e.g., ITS/ETS + NLC) significantly increases computational complexity, potentially causing analysis failures or incomplete optimizations.
- 2) Impact of data heterogeneity on analysis. The differing evolutionary rates between ITS/ETS (ribosomal DNA) and NLC (nuclear genes) may introduce biases in branch length estimation, requiring partitioned models for correction.
- 3) Low support for certain taxa. Taxa relying on only a few genes (e.g., ITS/ETS) may have unstable placements due to data sparsity, necessitating shared markers to improve accuracy.
- 4) Limitations on species-tree methods. Non-random distribution of missing data (e.g., incomplete NLC coverage across taxa) may reduce the reliability of multispecies.

As demonstrated by Portik et al. (2023 *Mol. Biol. Evol.* 40, msad109), the gigamatrix approach enables reliable species placement using ITS/ETS data. We therefore reconstructed maximum likelihood (ML) trees (treating each marker as a single partition) to maximize sampling in *Artemisia*. While limitations exist, this strategy balances phylogenetic resolution with taxonomic coverage. These methodological considerations are detailed in Methods (Lines 476–482) and Results (Lines 135–138), enhancing technical transparency.

-I was surprised to find no reference to data availability along the manuscript regarding the resulting trees and associated metadata, which I believe should be also made available if the ultimate aim is for the scientific community to use in the future the resources generated in *Artemisia*. Based on that, the authors should add a statement as to whether editable versions of the megamatrix datasets and trees are, or will be, deposited in open repositories.

Response 5: We appreciate the reviewer's emphasis on data accessibility. The revised manuscript now includes a comprehensive data availability statement to ensure transparency:

- Sequence data: All nuclear, plastid, and ribosomal DNA sequences are deposited in GenBank, with accession numbers listed in Supplementary Data 2.

- Analytical resources: Multi-sequence alignment matrices, partition files, and phylogenetic trees are openly accessible via FigShare (<https://doi.org/10.6084/m9.figshare.28164335>).

- Taxonomic and distributional data: Standardized species names (aligned to POWO, GCC, WFO and COL) and specimen-based distribution data are provided in Supplementary Data 6 and Supplementary Data 2.

- Morphological datasets: Character matrices for sampled species are detailed in Supplementary Data 4 and 5, facilitating reproducibility and future taxonomic studies.

These resources are documented in Methods (Lines 598–608), supporting open science and future *Artemisia* research.

-The authors claimed that besides the NLC dataset (Low copy genes from genome skimming), they also built a NR dataset (ITS+ETS extracted from genome skimming), plus an additional dataset with ITS+ETS data from Genbank (Table S2). Fig. S1 (concatenated ML tree of NR data) only includes data from the NR dataset (i.e. genome skimming), and I understand that Fig. S9. (concatenated ML tree of the NR Data) includes data from the NR dataset and Genbank, however, these two files have the same title, yet the Fig. S9 is not cited anywhere in the main manuscript (unless I have missed it). This creates some sort of confusion to the reader. For example, in page 9 line 197-99 reads "2) In the plastome trees, seven out of the eight subgenera of *Artemisia* recognized in the NLC tree are not monophyletic, except for *A. subg. Pacifica*, for which a single species was sampled (Fig. 2a)." I understand that in the NLC only one species of subg. *Pacifica* (*A. chinensis*) has been sequenced, but ITS and ETS data is available for the other 3 species in the subgenus, as shown in Fig. 3 and Fig. S9, therefore the monophyly is confirmed with more than one species. This point could be further explained in the manuscript, and perhaps merge Fig. S1 and S9 into one, since both represent ITS/ETS data regardless of their origin (genbank vs. skimming)

Response 6: We appreciate the opportunity to address these concerns. We have implemented the following corrections:

1) Figure title standardization:

- Figure S1 (now Figure S2 in the revised manuscript): Now labeled as "Concatenated ML tree of 314 species (298 *Artemisia* + 16 allied species) based on 202 nuclear low-copy genes (NLC dataset) ".

- Figure S9 (now Figure S4 in the revised manuscript): Now labeled as "Concatenated ML tree of 414 species (394 *Artemisia* + 20 allied species) based on ribosomal

nuclear DNA sequences (ITS + ETS; NR dataset)", from both genome-skimming and GenBank sources.

This resolves dataset abbreviation ambiguities and ensures NLC/NR distinction.

2) Monophyly of *Artemisia* subg. *Pacifica*:

With newly generated genome-skimming data in our revised manuscript, all four species in this subgenus are now included in the analysis. Both low-copy nuclear genes (NLC dataset) and plastome dataset strongly supports its monophyly (see details in Figs. 2, 3, S2 and S6).

-In page 17, line 400-402, "For instance, *A. magellanica*, a newly sampled Patagonian species previously assigned to *A. subg. Artemisia* (ref. 77), was revealed to belong to *A. subg. Pectinatae* (Fig. 3)" After careful reading of the cited study (Ref. 77, Garcia et al. 2011) I am convinced that it represents a secondary reference, since the sequence data on *A. magellanica* used comes from an original study on focused on South American *Artemisa* by Pellicer et al. (2010), and therefore the following citation should be added instead of Ref. 77. (Pellicer,J., T.Garnatje, J. Molero, F. Pustahija, S. Siljak-Yakovlev, and J. Vallès.2010b. Origin and evolution of the South American endemic *Artemisia* species (Asteraceae): Evidence from molecular phylogeny, ribosomal DNA and genome size data. Australian Journal of Botany 58: 605–616.)

Response 7: Thank you for your careful review. The citation for the taxonomic history of *A. magellanica* has been updated from García et al. (2011) to the primary source (Pellicer et al., 2010b; Line 176), which provides direct molecular evidence for its placement. We have reviewed all references in the main text and supplementary materials to ensure primary sources are used throughout. We appreciate you prompting this important correction.

Thank you for your critical review and comments.

(End of document)

Response to Reviewers

Reviewer #1 (Remarks to the Author):

Upon review of the revised manuscript and rebuttal, I am satisfied with the changes and improvements made by the authors in response to my comments. The authors provided detailed rebuttals to each of my questions and made appropriate corrections to the text, figures, and tables as suggested, including supplemental files. As stated in my initial review, I consider this study a valuable contribution to advancing the understanding of this large, cosmopolitan, and complex genus, and I see value in applying the approach to other large and taxonomically-complex groups of organisms. The resulting global phylogeny is the most comprehensive to date for *Artemisia* and its allied genera. I also agree with the authors in that their taxonomic classification, although provisional as classifications tend to be, is insightful and will serve as a helpful framework for more detailed studies at the species level going forward.

Response 1: Thank you very much for your positive comments.

Nevertheless, I found one minor correction that should be made regarding the color codes in Figure S6. The color code for *Artemisia* subg. *Pacifica* is "orange" in the legend and elsewhere in the paper, but the lines in the phylogeny for these taxa are black in Fig. S6. I made this observation while looking over comments from Reviewer #4 in the author's rebuttal. Otherwise, I recommend acceptance of the revised manuscript.

Response 2: As suggested, we have corrected the color code for *Artemisia* subg. *Pacifica* in Figure S6 (now designated as Supplementary Figure 6 in the Supplementary Information). Additionally, we have thoroughly reviewed all other figures and their legends in the manuscript to ensure consistency. Thank you very much for your careful observation and valuable feedback.

Reviewer #1 (Remarks on code availability):

I looked over the files using the provided link, but I did not try to install or run any programs. I also went through the list of codes and opened them to check for access, and I didn't find any noticeable issues. The data, codes, phylogenies, etc all appeared to be available and accessible to anyone who would want to use them.

Response 3: Thank you for verifying the accessibility of our code and data. We have ensured open access to all research data and code to facilitate the replication and extension of our work.

Reviewer #2 (Remarks to the Author):

The manuscript has been carefully revised and I am content with the changes made or the explanations given by the authors when they were of different opinion. In my view it is fit for publication now.

Response 1: We appreciate your positive comments and recommendation for publication.

Reviewer #3 (Remarks to the Author):

The authors have addressed all of my comments with great attention to detail and have made several substantial changes to the manuscript in response. I have no further suggestions and congratulate the authors to their work!

Dr Benedikt Kuhnhäuser

Royal Botanic Gardens, Kew

Response 1: We appreciate your recognition and rigorous assessment. Your expert comments were instrumental in revising the manuscript, and we are grateful for your dedication to enhancing this work.

Reviewer #3 (Remarks on code availability):

I tried to access the provided link under "Code availability" (<https://doi.org/10.6084/m9.figshare.28164335>) but was directed to a page displaying "Error: DOI not found". Maybe the DOI is not active yet? As such, I have not been able to look at the code.

Beyond this, however, the paper largely uses publicly available software and describes the methodology in sufficient detail, so that I am satisfied that the methodology is generally transparent and reproducible.

Response 2: We appreciate your note regarding the Figshare DOI. This DOI will activate upon publication; until then, the code is accessible via the private Figshare link in the submission system: [<https://figshare.com/s/db64a399e1c9aafcc884>]. The code was also successfully uploaded to Code Ocean (July 15, 2025) and will be publicly executable post-publication: [<https://codeocean.com/capsule/4896089/tree>]. All accession codes in the Data Availability section have been updated with active DOI hyperlinks.

In the revised version, we further refined the manuscript per editorial requests (detailed in the revised Author Checklist).

(End of document)